Corrected: Author correction

# GWAS for male-pattern baldness identifies 71 susceptibility loci explaining 38% of the risk

Nicola Pirastu [1], Peter K. Joshi [1], Paul S. de Vries[2], Marilyn C. Cornelis[3], Paul M. McKeigue[4], NaNa Keum[5,6], Nora Franceschini[7], Marco Colombo[4], Edward L. Giovannucci[6,8,9], Athina Spiliopoulou[4,10], Lude Franke [11], Kari E. North[7], Peter Kraft[12], Alanna C. Morrison[2], Tõnu Esko [13,14] & James F. Wilson [1,15]

Male pattern baldness (MPB) or androgenetic alopecia is one of the most common conditions affecting men, reaching a prevalence of ~50% by the age of 50; however, the known genes explain little of the heritability. Here, we present the results of a genome-wide association study including more than 70,000 men, identifying 71 independently replicated loci, of which 30 are novel. These loci explain 38% of the risk, suggesting that MPB is less genetically complex than other complex traits. We show that many of these loci contain genes that are relevant to the pathology and highlight pathways and functions underlying baldness. Finally, despite only showing genome-wide genetic correlation with height, pathway-specific genetic correlations are significant for traits including lifespan and cancer. Our study not only greatly increases the number of MPB loci, illuminating the genetic architecture, but also provides a new approach to disentangling the shared biological pathways underlying complex diseases.

[1] Centre for Global Health Research, Usher Institute of Population Health Sciences and Informatics, University of Edinburgh, Teviot Place, Edinburgh EH8 9AG, Scotland. [2] Human Genetics Center, Department of Epidemiology, Human Genetics and Environmental Sciences, School of Public Health, The University of Texas Health Science Center at Houston, Houston, TX 77030, USA. [3] Department of Preventive Medicine, Northwestern University Feinberg School of Medicine, Chicago, IL 60611, USA. [4] Centre for Population Health Sciences, Usher Institute of Population Health Sciences and Informatics, University of Edinburgh, Teviot Place, Edinburgh EH8 9AG, Scotland. [5] Department of Food Science and Biotechnology, Dongguk University, Goyang, South Korea. [6] Department of Nutrition, Harvard T. H. Chan School of Public Health, Boston, MA 02115, USA. [7] Department of Epidemiology and Carolina Center for Genome Sciences, University of North Carolina, Chapel Hill, NC 27599, USA. [8] Department of Epidemiology, Harvard T. H. Chan School of Public Health, Boston, MA 0211, USA. [9] Channing Division of Network Medicine, Department of Medicine, Brigham and Women's Hospital and Harvard Medical School, Boston, MA 02115, USA. [10] Pharmatics Ltd., Edinburgh EH16 4UX, Scotland. [11] Department of Genetics, University Medical Center, 9713 GZ Gröningen, The Netherlands. [12] Program in Genetic Epidemiology and Statistical Genetics, Harvard T. H. Chan School of Public Health, Boston, MA 02115, USA. [13] Estonian Genome Center, University of Tartu, 51010 Tartu, Estonia. [14] Broad Institute of Harvard and MIT, Cambridge, MA 02142, USA. [15] MRC Human Genetics Unit, Institute of Genetics and Molecular Medicine, University of Edinburgh, Western General Hospital, Crewe Road, Edinburgh EH4 2XU, Scotland. Correspondence and requests for materials should be addressed to N.P. (email: nicola.pirastu@ed.ac.uk)

Male pattern baldness (MPB) is a common multifactorial condition characterised by a progressive thinning of scalp hair with a very specific pattern. First, the hairline recedes at the temples (widow's peak) and then a bald patch develops on the crown, progressing such that eventually only a horseshoe of hair is left on the sides and back of the head (the Hippocratic wreath).

MPB co-exists with numerous other important pathologies such as cardio-metabolic diseases[1] and prostate cancer[2], suggesting a common underlying biology, and has negative impacts on body image and social perceptions[3].

The development of MBP is clearly linked to the response to androgens although many other pathways, such as *Wnt* and *TGF-beta*, come into play. Despite MPB's high heritability (~80%), after the first identification of the *AR/EDA2R* locus on the X chromosome[4,5] it took several years to identify additional loci, bringing the overall number to 12[6–8]. Recently, a larger study including 20,000 people has greatly increased the number of loci although independent replication was not demonstrated[9]. Studies investigating the genetic correlations of MBP with other traits/diseases have been inconclusive. Despite the identification of several loci associated with MBP, a number of questions remain unanswered. It is unclear if multiple SNPs at the same locus substantially contribute to MBP risk; moreover, although many predisposing loci have been identified, which genes are actually causal to MPB and which common pathways they share is not always clear; finally, despite numerous described co-morbidities with MPB, the biology responsible for these correlations remains largely unknown.

In order to shed light on these issues, we performed a large-scale genome-wide association (GWA) analysis to uncover the genetic architecture of baldness. We find 71 significantly associated loci, 30 of which were previously undescribed. These loci account for 38% of the heritability of MPB, suggesting a relatively low level of complexity in the genetic architecture. Further, we show that the identified loci contain genes which are enriched in pathways known to be important for hair follicle development and growth. Finally, we reveal that although MBP shows overall genetic correlation only with height, this changes once the correlations are estimated only over the enriched pathways, helping to understand the observed epidemiological results.

## Results

**Genome-wide association.** The discovery analysis was conducted using 25,662 MPB cases and 17,928 controls (see "Methods" for definition) from UK Biobank (UKB)[10], who self-identified as British and were also considered genomically British by UKB. All imputed SNPs on the autosomes and genotyped SNPs on chromosome X were used, for a total of 27,512,692. Discovery GWA identified 12,192 SNPs significantly ($p < 5 \times 10^{-8}$) associated with MBP. Replication was sought for significant SNPs using several independent cohorts: UKB participants of non-British origin, two prospective cohorts of European Americans (ARIC[11] and HPFS[12]) and 23andMe Inc.[13] (Supplementary Data 1 describes each cohort), for a total of 16,824 cases and 14,288 controls. We replicated 11,624 SNPs at $p < 0.05$, corresponding to 95.3% of the SNPs identified in discovery (Supplementary Data 2). SNPs were then divided into 71 independent loci (as defined in "Methods" and Table 1), hereafter referred to using numbers 1–71. Of the 71 loci, 30 were newly identified in this study and 41 loci had previously been described and replicated in association studies[9,14]. (Fig. 1; regional Manhattan plots of the associated loci can be seen in Supplementary Figs. 1–71.)

In order to understand if at each locus the association was due to a single SNP or if multiple association signals were present, we conducted GCTA-COJO analysis[15]. The 71 loci contain 107 distinct SNPs (Supplementary Data 3): 22 loci include from 2 up to 5 distinct associations (Table 1).

No heterogeneity of effect was detected between the genomically and non-genomically British sub-cohorts over the 107 SNPs (Fig. 2; the comparison of effects for each distinct SNP). Genomic heritability analysis using 20,000 random unrelated UKB samples reveals that 94% of variance on the liability scale could be attributed to genetic variation: 82% of heritability could be attributed to the autosomes and 12% could be attributed to the X chromosome. Our 107 genome-wide significant SNPs explain 38% of the total heritability; 32% of the autosomal and 73% of the X chromosome contributions. MBP is therefore probably one of the most heritable complex traits[16,17] and one for which we can explain a remarkably large part of heritability with relatively few SNPs compared to other complex traits[18] including autoimmune diseases[19].

**Functional annotation of discovered loci.** All distinct SNPs and their LD proxies were annotated with Haploreg v4.1[20]. This set contains only one coding SNP: rs17265513, a non-synonymous variant of *ZHX3*. Tissue-specific enhancer enrichment analysis highlights various tissues (Supplementary Data 4), the most significant of which are mesenchymal stem cell-derived chondrocyte cultured cells ($p = 3 \times 10^{-5}$) but also foreskin keratinocyte and melanocyte primary cells ($p = 1 \times 10^{-4}$, $2 \times 10^{-4}$). All SNPs with $p < 1 \times 10^{-5}$ in the 71 replicated loci were subject to tissue- and gene-set enrichment with DEPICT 1.1[21]. Tissue enrichment analysis did not yield any results (Supplementary Data 5), however gene-set enrichment revealed 202 enriched sets (Supplementary Data 6), most involving either morphogenesis or regulation of transcription. All associated loci contain genes except for two, which are in gene deserts; the total number of protein coding genes at these loci is 219 (Supplementary Data 7). In order to determine if these 219 genes clustered in known pathways, we ran enrichment analysis using ConsensusPathDB-human[22], which identified only nine significant enriched pathways (Supplementary Data 8). We thus performed custom prioritisation (see "Methods"), which allowed us to select 72 genes in 60 loci (Table 1 and Supplementary Data 9). Eleven of the selected genes also encode druggable targets according to DGIdb 2.0[23] (Supplementary Data 10).

Androgen receptor signalling (see Supplementary Data 11 for the description of the pathways included in the list) is implicated by seven genes at six loci. The *AR/EDA2R* locus on chromosome X shows the strongest association (rs4827528; $p < 1 \times 10^{-350}$; OR, 3.4), and three distinct signals. *SRD5A2* encodes a protein that converts testosterone to dihydrotestosterone (DHT), with five distinct hits near the gene. Additionally, our SNP rs5934505, located on chromosome X close to *FAM9B*, has also been associated with testosterone levels[24].

The selected genes were tested for enrichment of known pathways and gene sets using ConsensusPathDB-human[22]. Many pathways known to affect MBP were enriched, in particular *Wnt* signalling and apoptosis (Supplementary Data 12). Three hundred and three gene ontology (GO) terms were significant, mostly related to the regulation of developmental processes and morphogenesis (Supplementary Data 13). In order to identify possible subgroups of the genes which map to known pathways, we first built an adjacency matrix based on the co-membership of each gene in a specific pathway, then carried out community detection on this network (see "Methods").

Three main groups were found: genes linked to *Wnt* signalling (*LGR4*, *RSPO2*, *WNT3*, *WNT10A*, *SOX13*, *DKK2*, *TWIST1*, *TWIST2*, *IQGAP1* and *PRKD1*), genes involved in apoptosis (*BCL2*, *DFFA*, *TOP1*, *IRF4* and *MAPT*) and a third more heterogeneous group including the androgen receptor and *TGF-*

**Table 1 Summary of MPB loci**

| Locus # | Chr | Start | End | Replicated | N distinct SNPs | N significant SNPs | Selected genes | Most significant p |
|---|---|---|---|---|---|---|---|---|
| 1 | 1 | 10509603 | 11264064 | Y | 2 | 168 | *DFFA* | 6.04E−49 |
| 2 | 1 | 24570475 | 25511358 | Y | 4 | 143 | *SYF2-RUNX3* | 1.28E−20 |
| 3 | 1 | 41354272 | 41407656 | N | 1 | 7 | *CITED4* | 5.74E−09 |
| 4 | 1 | 47917547 | 48002096 | Y | 1 | 26 | *FOXD2* | 6.57E−12 |
| 5 | 1 | 50019849 | 51742123 | N | 1 | 248 | *DMRTA2* | 6.47E−13 |
| 6 | 1 | 118536585 | 119857264 | Y | 4 | 268 | *WARS2* | 3.14E−19 |
| 7 | 1 | 151997199 | 152310892 | N | 1 | 14 | *RPTN-TCHH* | 1.08E−10 |
| 8 | 1 | 170194002 | 170907900 | Y | 2 | 572 | *PRRX1* | 2.78E−20 |
| 9 | 1 | 203714162 | 203979114 | N | 1 | 6 | *SOX13* | 8.57E−15 |
| 10 | 2 | 6089096 | 6573742 | Y | 1 | 16 | *ESPN(trans-eQTL)* | 8.24E−10 |
| 11 | 2 | 30585541 | 30632740 | N | 1 | 16 | *LCLAT1* | 7.66E−09 |
| 12 | 2 | 31490493 | 33520716 | Y | 5 | 207 | *SRD5A2* | 1.90E−26 |
| 13 | 2 | 60517707 | 60697464 | N | 1 | 12 | | 8.77E−09 |
| 14 | 2 | 67675400 | 68172629 | Y | 1 | 54 | | 7.75E−12 |
| 15 | 2 | 70196019 | 70558994 | N | 1 | 116 | *FAM136A* | 4.05E−08 |
| 16 | 2 | 145638766 | 145881714 | N | 1 | 12 | | 1.90E−08 |
| 17 | 2 | 174268562 | 174617984 | N | 1 | 5 | *CDCA7* | 4.61E−10 |
| 18 | 2 | 176794997 | 177889303 | Y | 2 | 232 | *HOXD3* | 1.49E−14 |
| 19 | 2 | 219291486 | 219854317 | Y | 1 | 82 | *WNT10A* | 1.07E−19 |
| 20 | 2 | 222988705 | 223108422 | N | 1 | 23 | *PAX3* | 3.63E−10 |
| 21 | 2 | 239495665 | 239950595 | Y | 2 | 171 | *TWIST2* | 1.43E−39 |
| 22 | 3 | 107181737 | 107427969 | N | 1 | 1 | *BBX* | 1.18E−10 |
| 23 | 3 | 125899126 | 126278765 | Y | 2 | 26 | | 4.84E−13 |
| 24 | 3 | 138648310 | 139032333 | Y | 2 | 59 | *COPB2* | 2.26E−21 |
| 25 | 3 | 141093285 | 141336557 | N | 1 | 9 | *ALPL(trans-eQTL)* | 2.75E−08 |
| 26 | 3 | 151406296 | 151781794 | Y | 2 | 252 | *AADAC* | 8.21E−18 |
| 27 | 4 | 81095299 | 81307025 | Y | 1 | 108 | *FGF5-PRDM8* | 4.60E−25 |
| 28 | 4 | 106008586 | 106038169 | N | 1 | 11 | *TET2* | 2.63E−09 |
| 29 | 4 | 107146797 | 108477318 | Y | 2 | 47 | *DKK2* | 8.10E−11 |
| 30 | 5 | 122084693 | 122490489 | N | 1 | 6 | *PPIC-PRDM6* | 1.44E−08 |
| 31 | 5 | 157510608 | 158614607 | Y | 1 | 206 | *EBF1-UBLCP1* | 2.29E−44 |
| 32 | 6 | 226393 | 615736 | Y | 1 | 33 | *IRF4* | 2.35E−52 |
| 33 | 6 | 8915298 | 10241822 | Y | 3 | 452 | *OFCC1* | 9.05E−29 |
| 34 | 6 | 44684124 | 45756667 | N | 1 | 1 | *RUNX2* | 3.62E−08 |
| 35 | 6 | 105957344 | 106256325 | N | 2 | 32 | | 9.09E−14 |
| 36 | 6 | 126268105 | 127212478 | Y | 1 | 359 | *CENPW* | 1.66E−17 |
| 37 | 7 | 428825 | 574149 | N | 2 | 109 | *PDGFA* | 6.00E−20 |
| 38 | 7 | 18683672 | 19141259 | Y | 2 | 132 | *TWIST1* | 7.34E−63 |
| 39 | 7 | 46798132 | 47074382 | Y | 1 | 223 | *EPS15P1* | 1.07E−17 |
| 40 | 7 | 68566391 | 69822973 | Y | 2 | 531 | *AUTS2* | 2.06E−41 |
| 41 | 7 | 130742066 | 131010943 | N | 1 | 11 | | 2.94E−08 |
| 42 | 8 | 108420980 | 110628938 | Y | 5 | 298 | *RSPO2* | 2.13E−22 |
| 43 | 8 | 116416322 | 117271922 | N | 2 | 143 | *TRPS1* | 2.71E−10 |
| 44 | 9 | 109559841 | 109695139 | N | 1 | 4 | *ZNF462* | 7.25E−09 |
| 45 | 10 | 62826952 | 62995640 | N | 1 | 18 | *RHOBTB1* | 4.59E−09 |
| 46 | 10 | 78076367 | 78640827 | Y | 1 | 662 | *C10orf11* | 9.31E−24 |
| 47 | 10 | 126266230 | 126576345 | Y | 1 | 66 | *METTL10-FAM53B* | 2.58E−17 |
| 48 | 11 | 27360070 | 27567092 | N | 1 | 31 | *LGR4* | 8.99E−12 |
| 49 | 11 | 44374074 | 44526229 | Y | 1 | 94 | *ALX4* | 2.14E−19 |
| 50 | 12 | 26356406 | 26458670 | Y | 1 | 22 | *SSPN* | 1.62E−16 |
| 51 | 12 | 27983266 | 28116111 | Y | 1 | 18 | *PTHLH* | 3.43E−09 |
| 52 | 12 | 28655747 | 29440112 | N | 1 | 208 | *FAR2* | 5.73E−14 |
| 53 | 12 | 130556647 | 130579675 | Y | 1 | 14 | | 1.73E−12 |
| 54 | 14 | 30548135 | 30575473 | N | 1 | 2 | *PRKD1* | 3.65E−08 |
| 55 | 15 | 56926063 | 57873575 | Y | 1 | 95 | | 1.08E−11 |
| 56 | 15 | 69964858 | 70048984 | Y | 1 | 93 | | 4.25E−15 |
| 57 | 15 | 90466945 | 91088596 | N | 1 | 49 | *CRTC3-IQGAP1* | 1.90E−08 |
| 58 | 16 | 14377400 | 14406119 | Y | 1 | 12 | | 1.42E−09 |
| 59 | 17 | 12250947 | 12521291 | N | 1 | 23 | | 1.54E−09 |
| 60 | 17 | 43074095 | 44865603 | Y | 3 | 3920 | *WNT3-MAPT-PLEKHM1* | 1.49E−26 |
| 61 | 17 | 55217723 | 55287871 | Y | 1 | 88 | *MSI2* | 8.95E−23 |
| 62 | 18 | 9971288 | 10451695 | N | 1 | 1 | *APCDD1* | 7.09E−10 |
| 63 | 18 | 42437417 | 42838477 | Y | 1 | 160 | *SETBP1* | 8.03E−25 |
| 64 | 18 | 60924613 | 60943706 | N | 1 | 3 | *BCL2* | 2.35E−08 |
| 65 | 20 | 21627103 | 22417667 | Y | 3 | 910 | *PAX1* | 1.42E−105 |
| 66 | 20 | 39620847 | 40270138 | Y | 1 | 96 | *ZHX3-TOP1* | 4.04E−16 |
| 67 | 20 | 55269324 | 55435053 | N | 2 | 4 | *TFAP2C* | 5.40E−11 |
| 68 | 21 | 36198190 | 36238517 | Y | 1 | 7 | *RUNX1* | 9.94E−19 |
| 69 | 21 | 45935196 | 46174999 | N | 1 | 20 | | 1.06E−09 |
| 70 | 23 | 8874087 | 8916646 | Y | 1 | 2 | *FAM9B* | 4.07E−13 |
| 71 | 23 | 55163863 | 67910139 | Y | 3 | 114 | *AR* | 1E−879 |

The table reports for each replicated locus the genomic position, if the locus had been previously described and replicated, the number of distinct SNPs, the total number of significant replicated SNPs, the prioritised genes and the most significant *p* value

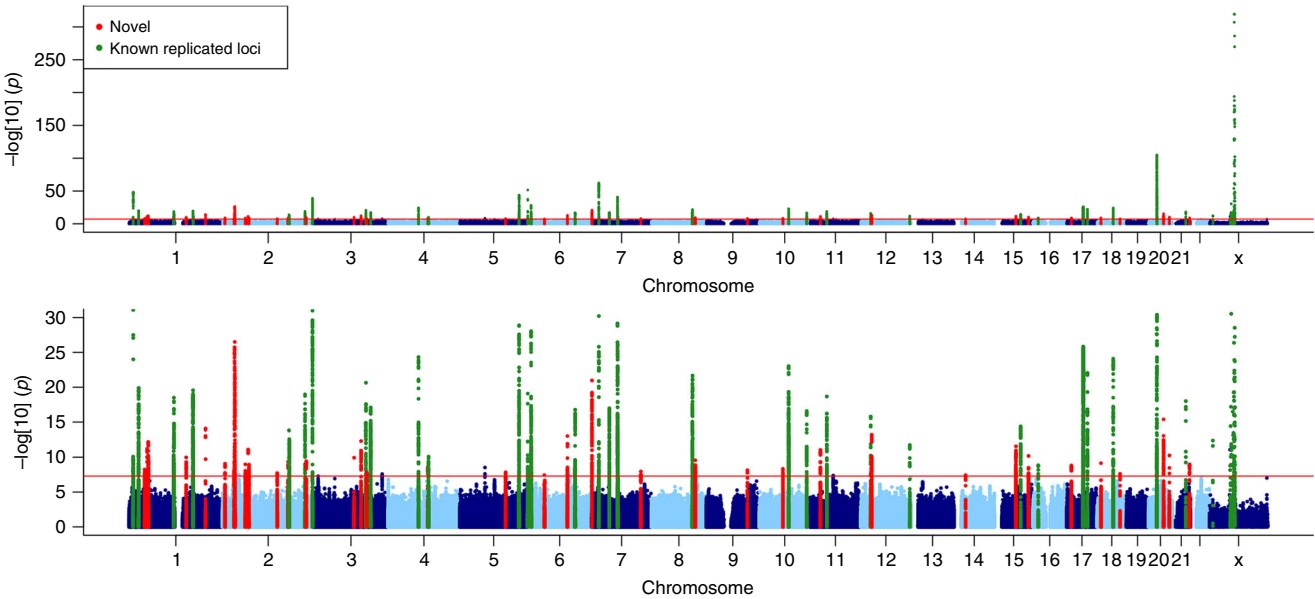

**Fig. 1** Manhattan plots of the discovery phase association analysis. In the lower panel, the $p$ values have been capped at $1 \times 10^{-30}$ to aid legibility. Loci have been coloured in red if novel, orange if reported but not replicated and green if known and replicated

beta pathways (*RUNX2*, *RUNX3*, *PTHLH*, *ALPL*, *AR*, *RUNX1*, *PDGFA*, *SRD5A2*, *FGF5* and *PAX3*). Supplementary Fig. 1 shows the membership of each gene in each pathway. Although many different pathways have been implicated in the development of MBP[25], our results suggest that in addition to the androgen receptor pathway, for which we confirm a prominent function, the *Wnt* and apoptosis pathways play central roles. MPB is characterised by a shorter growth (anagen) phase, which has been associated with increased apoptosis[25] of the hair follicle cells. These results suggest that the anagen phase becomes shorter because of differences in the genes regulating apoptosis. The *Wnt* pathway has been implicated in the transition from the resting (telogen) phase to the anagen phase[26], and also in the determination of the fate of the stem cells in the hair bulge[27], which are both dysregulated in balding tissue.

A second community analysis was conducted based on the co-membership of the 303-specific GO terms. In this case, only two groups of genes were detected (Supplementary Data 14): the first characterised by genes enriched in 106 different known pathways mostly linked to signalling (Supplementary Data 15), while the second group featured genes enriched in 12 known pathways related mostly to apoptosis and development (Supplementary Data 16). Both the "signalling" and "other" communities were enriched for GO terms related to the regulation of developmental processes. Given these results, we used STRING[28] to build an interaction network between the prioritised genes. Thirty-three of the selected genes were connected in a large network which included 8 of the 13 genes differentially expressed in balding dermal papilla cells under DHT stimulation[29]. Overlaying the GO term-based communities with the interaction pathways shows that the genes in the signalling community (Supplementary Fig. 2) are located at the centre of the network and show a high degree of interconnectivity, suggesting considerable interaction amongst the different regulatory signalling systems involved in hair growth. The genes in the other community show less interconnectivity, consistent with their not being in similar pathways. These results suggest that while the genes in the "signalling" community receive the various signals regulating hair growth, they then interact with those in the "other" community to transduce these signals into responses. These results could also be

due to a difference in knowledge between the genes in the two communities, however when we compare the per-gene number of associated GO terms in each community, we actually detected a higher median and mean number of associated terms to the second community genes (median, 65.5 vs. 40; and mean, 64 vs. 59.1) suggesting that this is not the case for our results.

**Locus- and pathway-specific pleiotropy**. Pleiotropy analysis conducted with GENOSCORES (https://pm2.phs.ed.ac.uk/genoscores/) revealed that 14 loci show strong correlation with previously published GWAS loci including various diseases and quantitative traits (Supplementary Data 17). Genetic correlation analysis using LD-hub[30,31] did not yield any results (Supplementary Data 18); we thus verified the overlap between the predisposition to baldness and other traits using polygenic risk scores, PGRS[32]. Generally, the genetic overlap between traits is sought either for a single variant (pleiotropy) or at the genome-wide level (genetic correlation), however it is plausible that any genetic overlap is limited to specific pathways and that this correlation is lost in the overall score. We thus tested each of the enriched pathways and the three pathway-based communities. Each PGRS was tested against numerous traits on the genomically British UKB cohort (see "Methods" for details). Table 2 describes the SNPs used to create each locus score and which loci were attributed to each pathway. The overall score was significant only for height, with higher risk of baldness associated with lower stature (Fig. 3; also see Supplementary Data 19 for the significant results and Supplementary Data 20 full results), indeed seven individual loci independently show this same effect. For other traits, associations in different directions cancel out and so the overall score shows no evidence of an effect. Looking at pathway-specific scores, we see that predisposition to baldness is associated with lower male lifespan, when considering only several pathways linked to *Wnt* signalling, whereas this genetic predisposition is shared with increased risk of any cancer, when considering only variants in the apoptosis pathway or community. Although in both examples, one of the loci in the pathway is significant by itself, the pathway score was stronger, suggesting the other genes in the pathway also contribute, with the same direction of effect.

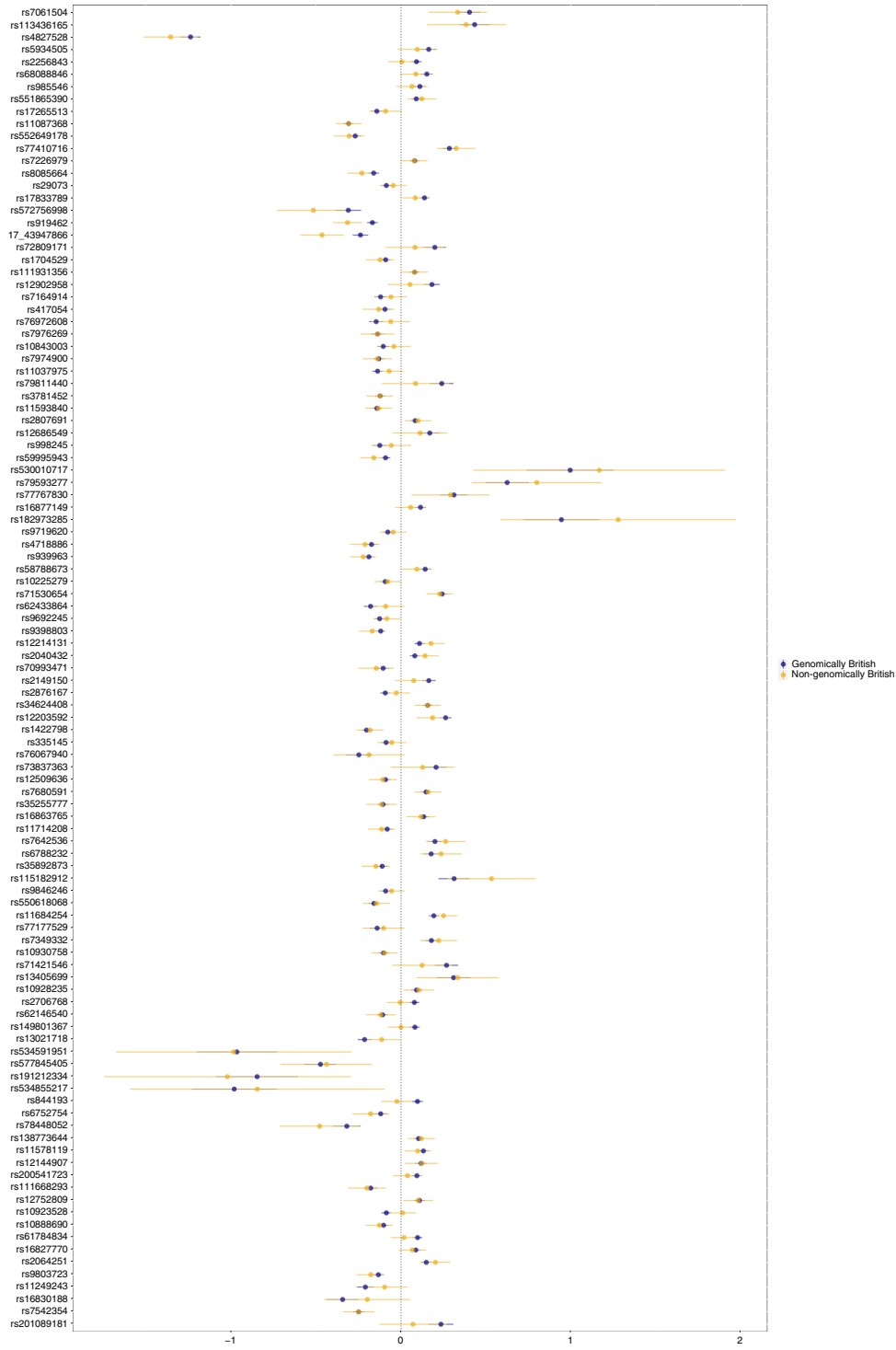

**Fig. 2** Forest plot of SNP effects in genomically and non-genomically British cohorts. The figure reports the effect and confidence intervals for each distinct SNP in the genomically British (in blue) and non-genomically British (in yellow). None of the SNPs showed significant differences between the two cohorts

Finally, baldness risk loci in the WNT ligand biogenesis and trafficking and Class B/2 (Secretin family receptors) pathways were also associated with height, despite none of the individual loci in these pathways being significant: this suggests a "pathway-wide" effect. Therefore, baldness shows pathway-specific genetic correlations, which provide a potential biological basis to observed epidemiological correlations. Pathway-specific genetic correlations hold promise in disentangling the shared biological pathways underpinning complex diseases.

## Discussion

In this study, we have identified and replicated numerous new loci predisposing to MPB which, together with the previously identified ones, can explain a large proportion of the estimated heritability. Furthermore, we have shown that for many of them, multiple distinct SNPs participate in determining MBP risk. We have also shown that MBP loci contain genes attributable to specific pathways, some of which predispose to other traits such as longevity and cancer.

**Table 2 Pathways used to generate pathway polygenic risk scores**

| Pathway | Loci in pathway | N SNPs | SNP list |
|---|---|---|---|
| Apoptosis | 1;32;64 | 4 | rs201089181;rs7542354;rs12203592;rs7226979 |
| Apoptotic signalling pathway | 1;32;64 | 4 | rs201089181;rs7542354;rs12203592;rs7226979 |
| Integrated pancreatic cancer pathway | 1;37;64;66 | 6 | rs201089181;rs7542354;rs9692245;rs62433864;rs7226979;rs17265513 |
| Apoptosis modulation and signalling | 1;54;64 | 4 | rs201089181;rs7542354;rs417054;rs7226979 |
| HIV-1 Nef: Negative effector of Fas and TNF-alpha | 1;64 | 3 | rs201089181;rs7542354;rs7226979 |
| Caspase cascade in apoptosis | 1;64;66 | 4 | rs201089181;rs7542354;rs7226979;rs17265513 |
| Prostate cancer—Homo sapiens (human) | 12;37;64;71 | 11 | rs534855217;rs191212334;rs577845405;rs534591951;rs13021718;rs9692245;rs62433864;rs7226979;rs4827528;rs113436165;rs7061504 |
| Proteoglycans in cancer—Homo sapiens (human) | 19;21;38;57;60 | 9 | rs7349332;rs11684254;rs550618068;rs71530654;rs10225279;rs111931356;17_43947866;rs919462;rs572756998 |
| ESC pluripotency pathways | 19;27;37;60 | 7 | rs7349332;rs7680591;rs9692245;rs62433864;17_43947866;rs919462;rs572756998 |
| Pathways in cancer—Homo sapiens (human) | 19;27;37;60;64;68;71 | 12 | rs7349332;rs7680591;rs9692245;rs62433864;17_43947866;rs919462;rs572756998;rs7226979;rs68088846;rs4827528;rs113436165;rs7061504 |
| Class B/2 (Secretin family receptors) | 19;51;60 | 5 | rs7349332;rs10843003;17_43947866;rs919462;rs572756998 |
| Wnt signalling pathway | 19;54;60 | 5 | rs7349332;rs417054;17_43947866;rs919462;rs572756998 |
| Wnt signalling pathway and pluripotency | 19;54;60 | 5 | rs7349332;rs417054;17_43947866;rs919462;rs572756998 |
| WNT ligand biogenesis and trafficking | 19;60 | 4 | rs7349332;17_43947866;rs919462;rs572756998 |
| Wnt signalling in kidney disease | 19;60 | 4 | rs7349332;17_43947866;rs919462;rs572756998 |
| DNA damage response (only ATM dependent) | 19;60;64 | 5 | rs7349332;17_43947866;rs919462;rs572756998;rs7226979 |
| Endochondral ossification | 2;25;34;51 | 7 | rs16830188;rs11249243;rs9803723;rs2064251;rs11714208;rs70993471;rs10843003 |
| Regulation of nuclear SMAD2/3 signalling | 2;34;68;71 | 9 | rs16830188;rs11249243;rs9803723;rs2064251;rs70993471;rs68088846;rs4827528;rs113436165;rs7061504 |
| TGF_beta_Receptor | 2;34;68;71 | 9 | rs16830188;rs11249243;rs9803723;rs2064251;rs70993471;rs68088846;rs4827528;rs113436165;rs7061504 |
| Transcriptional misregulation in cancer—Homo sapiens (human) | 20;34;37;68 | 5 | rs77177529;rs70993471;rs9692245;rs62433864;rs68088846 |
| Activation of the TFAP2 (AP-2) family of transcription factors | 3;67 | 3 | rs16827770;rs551865390;rs985546 |
| White fat cell differentiation | 31;32 | 2 | rs1422798;rs12203592 |
| Development of pulmonary dendritic cells and macrophage subsets | 32;34 | 2 | rs12203592;rs70993471 |
| Notch-mediated HES/HEY network | 34;38;71 | 6 | rs70993471;rs71530654;rs10225279;rs4827528;rs113436165;rs7061504 |
| Regulation of FZD by ubiquitination | 42;48 | 6 | rs182973285;rs16877149;rs77767830;rs79593277;rs530010717;rs79811440 |
| Regulation of apoptosis by parathyroid hormone-related protein | 51;64 | 2 | rs10843003;rs7226979 |
| Signalling by Wnt | 9;19;29;42;48;60 | 13 | rs78448052;rs7349332;rs73837363;rs76067940;rs182973285;rs16877149;rs77767830;rs79593277;rs530010717;rs79811440;17_43947866;rs919462;rs572756998 |
| TCF-dependent signalling in response to WNT | 9;29;42;48;60 | 12 | rs78448052;rs73837363;rs76067940;rs182973285;rs16877149;rs77767830;rs79593277;rs530010717;rs79811440;17_43947866;rs919462;rs572756998 |

The first column represents the name of the pathway, the second the loci which contain genes relative to the pathway (i.e., 1;32;64 means locus 1, locus 32 and locus 64), the third the number of distinct SNPs in the loci, and the fourth the names of the SNPs

Compared with the recent GWA study by Heilmann et al.[9] describing 63 loci, we were unable to replicate 17 at genome-wide significance, while the remaining 46 can be summarised into 41 of ours due to the different locus definition. We were thus able to discover and replicate 30 novel loci, almost doubling the number known. The differences in loci discovered are probably partly due to their focus on early onset MBP and to our much greater sample size, which gave us an increased power to detect variants of smaller effect.

Recently, Haagenars et al.[33] used the same UK Biobank data to create a prediction score for MBP. As they were making predictions only, they did not seek replication and used LD clumping to identify separate predictive SNPs. Here, we found that of 287 loci reported in their work, 3 could not be replicated in our analyses, 18 had $p$ values over the significance threshold in our discovery analysis while the remaining 266 could be summarised by 106 of the SNPs selected in this study. Finally, they fail to

detect locus 34 on chromosome 6, which contains *RUNX2*. These differences are probably due to the different phenotype definition in the two studies and to the different nature of the study design.

For the majority of the identified loci, it was possible to select a convincing candidate through the integration of numerous sources. This is particularly important in order to understand the underlying biology of MBP and to identify new targets for therapies. In this regard, the fact that for 11 of the selected genes, a drug already exists opens several new possibilities for the treatment of MBP.

One of the most interesting contributions of our work is the development of a new way to analyse pleiotropy neither at a genome-wide level nor limited to single loci, but rather looking at specific pathways using scores estimated over them to elucidate pleiotropic correlations. This approach is important for two

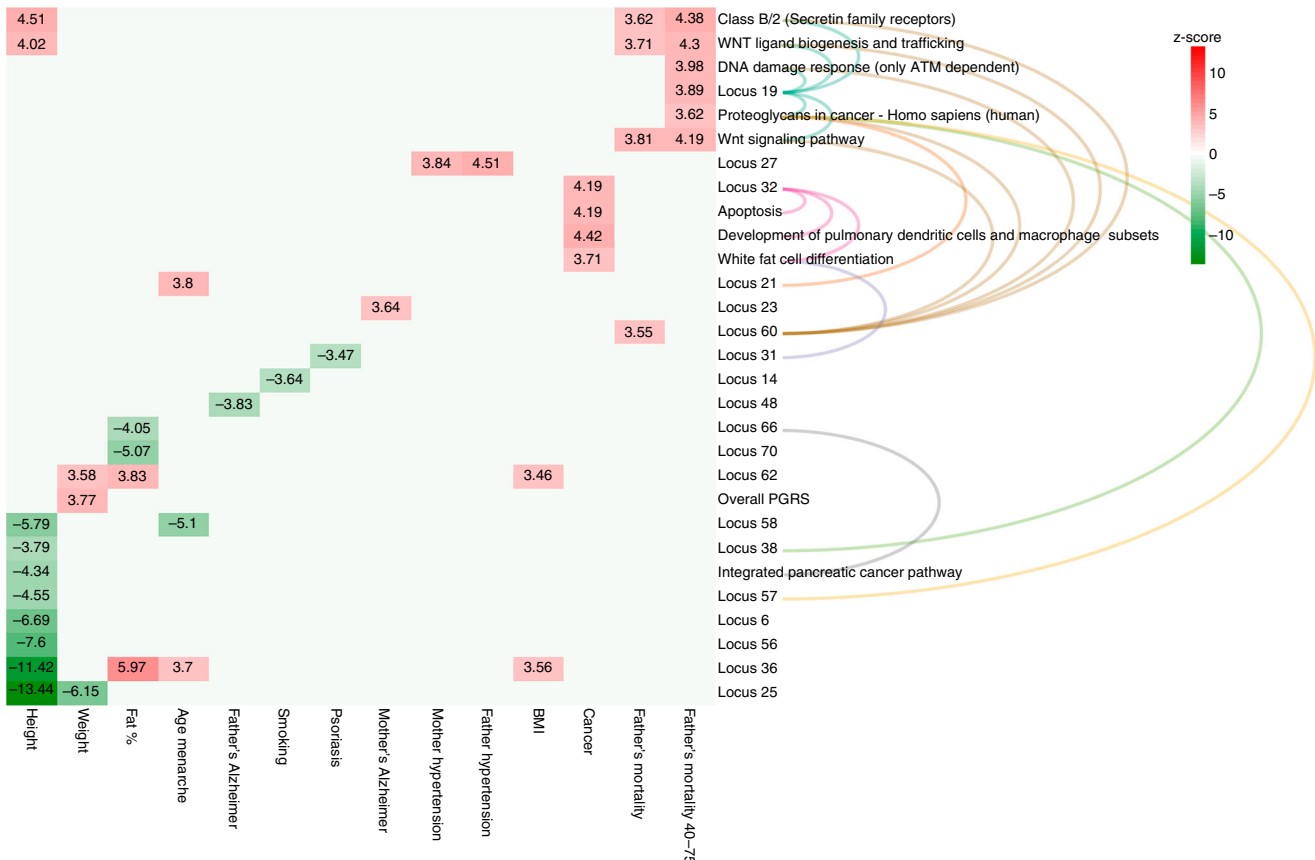

**Fig. 3** Significant effects of individual baldness loci and enriched pathways on disease or mortality risk. Arcs between the loci and the pathways indicate the membership of the genes in that locus in that pathway. The colour and the numbers represent the z-scores from the regressions. Only significant results ($q$ value <0.05) have been reported. Although in several cases, the pathway effects seem to be mainly driven by specific loci, in most cases the overall effect is greater than that of any individual locus. Moreover, we can see examples of pathway-specific effects not attributable to any specific locus, for example, the effect of WNT ligand biogenesis and Class B/2 (Secretin family receptors) pathways on height is detected only at the pathway level and is in the opposite direction to the overall polygenic score and those of the individual loci

reasons, first because it suggests a potential biological mechanism for observed epidemiological correlations. Pathway-specific genetic correlations also show that we can use outcomes to weight the contribution of SNPs to the pathway functionality, thus creating an instrument to study the effect of specific pathways on diseases and traits even without genome-wide significance.

We have observed an $h^2_{SNP}$ of 94%, which suggests that MBP is more a polygenic trait than a complex one, with very little environmental component. This estimate is in accordance with previous studies which have estimated dichotomised MBP heritability to be 89% for clear-cut vertex balding and 96% for recessive balding[34]. Two recent papers[9,33] have estimated $h^2_{SNP}$ to be significantly lower (0.48 and 0.52), however we must consider that in the first case a different method of converting the explained phenotypic variance from the observed scale to the liability one was used instead of the one implemented in GCTA while in the other rather dubious baldness cases were included (group 2 on the scale used in the UKB study), which may explain the difference.

In conclusion, we show baldness to be one of the most heritable complex traits, for which we are able to explain nearly two fifths of the heritability with only 71 loci. Recurring themes in complex trait genetics are highlighted, including multiple distinct signals in many loci, aggregation of genetic effects in pathways important to the trait and widespread pleiotropy with other traits and diseases; but we also emphasise pathway-specific genetic correlations as a new approach to dissect the genetics and biology of complex diseases.

## Methods

**Study approval.** All subjects gave written informed consent. UK Biobank has approval from the North West Multi-centre Research Ethics Committee (MREC), In Scotland, UK Biobank has approval from the Community Health Index Advisory Group (CHIAG). ARIC has approval from the ARIC Publications Committee and the ARIC Coordinating Center. The HPFS was approved by the IRB of the Harvard School of Public Health.

**Phenotype definition and cohort description.** In UK Biobank samples, the degree of baldness was defined as part of a self-administered questionnaire. Each male participant was asked to recognise his hair/balding pattern comparing with four different pictures. The first would correspond to no sign of baldness, the second to a II–IIIa grade on the Norwood–Hamilton[35] scale, the third to a IIIv–IV grade and the fourth to a V+ grade. To be certain of using people actually affected by MPB, we used as cases people who self-identified themselves as being in the third or fourth group while as controls people in the first category. Given that people in UK Biobank have been selected to be of ages between 40 and 69, by when baldness is fully developed, no age filter was applied. Genotypes both measured and imputed were obtained from UK Biobank and no further QC was applied. Details can be found at http://www.ukbiobank.ac.uk/wp-content/uploads/2014/04/UKBiobank_genotyping_QC_documentation-web.pdf and http://www.ukbiobank.ac.uk/wp-content/uploads/2014/04/imputation_documentation_May2015.pdf.

Health Professionals Follow-up Study (HPFS) is an on-going prospective analysis of male US health professionals, information on hair pattern at age 45 was collected by self-administered questionnaires during the 1992 follow-up[36]. Participants were asked to select their hair pattern at age 45 from five images (no baldness, frontal-only baldness, frontal-plus-mild-vertex baldness, frontal-plus-moderate-vertex baldness and frontal-plus severe vertex baldness) modified from the Hamilton–Norwood scale[35] Cases were defined as experiencing frontal and (any) vertex baldness at age 45, which corresponds to at least a IV grade on the Hamilton–Norwood scale. Controls were defined as those experiencing no baldness at age 45.

The Atherosclerosis Risk in Communities Study (ARIC) is a prospective population-based cohort study of individuals from four US communities[11].

Baseline examinations took place from 1987 to 1989. During the fourth visit (1996–1998), baldness pattern was classified according to the Hamilton baldness scale, as modified by Norwood[35]. A trained technician in each clinic observed the participant's head from two angles (side and top), compared the natural hair pattern with a series of 12 figures, and chose the best matching figure. These figures corresponded to categories on the Hamilton baldness scale (I, II, IIa, III, IIIa, IIIva, III vertex, IV, V, Va, VI, VII). A score of 13 was recorded for complete baldness. More details about these measurements have been previously described[37]. Cases were defined as individual with grade III or higher, whereas controls were defined as individuals with grade I. Given that the participants of the ARIC study were all between 53 and 74 years old, no age filter was applied. We restricted our analysis to European-American men.

23andMe data were obtained from research participants of 23andMe, Inc.[9] who provided informed consent and participated in research online, under a protocol approved by the external AAHRPP-accredited IRB, Ethical & Independent Review Services. Participants provided responses to the "Hair Loss in Men and Women" survey. Only responses from male research participants were included. Survey questions included the following: "Please choose the image that best captures your hair's pattern and density. If your head is shaved, please answer for how your hair looks when grown out. If none of these images are similar to your hair's pattern and density, choose none of the above." (images a to s, corresponding to Hamilton scale); "Have you experienced hair loss or thinning?" (Yes, No, I'm not sure, Decline to state); "How old were you when you first started to notice hair loss?" (under age 18, 18–24, 25–29, 30–34, 35–39, 40–44, 45–49, 50–54, 55–59, 60–64, 65–69, 70–74, 75–79, 80 years or older, I'm not sure.) Cases reported having experienced hair loss or thinning, with onset before age 40, and current hair loss of Hamilton grade III or higher. Controls are at least 30 years old, and report not having experienced hair loss or thinning, and at most Hamilton grade I, or if age 50 or older, Hamilton grade II.

**Discovery association analysis on UK Biobank samples**. Case–control association analysis on UK Biobank genomically British samples was conducted in two steps. First genome-wide association using linear regression (which is equivalent to running the trend test) was conducted using RegScan[38] on all 27,512,692 imputed SNPs, which had a minor allele count of at least 100 using the following model:

$$\mathrm{male\_pattern\_baldness} \sim \mathrm{age} + \mathrm{pc.0} + \mathrm{pc.1} + \mathrm{pc.2} \\ + \ldots + \mathrm{pc.10} + \mathrm{array\,batch} + \mathrm{genotype}.$$

For chromosome X, given the unavailability of imputed SNPs, genotypes were used. All SNPs with $p < 1 \times 10^{-5}$ were extracted from the original bgen files using bgenix (http://www.well.ox.ac.uk/~gav/bgen_format/) and analysed using logistic regression using the same covariates as in the first step. For this step, SNPtest v2.5.2[39] was used. The two-step approach was used in order to make the analysis time tractable since available software would have taken months to run the logistic regression analysis even on a large computing cluster.

Potential $p$ value inflation was checked using genomic control and LD regression: $\lambda_{GC}$ 1.15, LD regression intercept 1.0451. The difference in value indicates that the inflation measured by genomic control is due to the GWA and not to stratification.

**Replication association analysis**. Replication association analysis was performed using four different cohorts: samples self-defined as British but genomically non-British from UK Biobank, HPFS and ARIC. Additionally, data from the recent MBP GWAS[13] was obtained from 23andMe. The association analyses in the replication cohorts were limited to only the 12,192 SNPs significant at the discovery step. After association analysis in each cohort, results were meta-analysed using the inverse variance approach as implemented in METAL[40]. Given the large sample size, no filter on MAF or imputation quality was applied. For UK Biobank non-genomically British samples, case–control association analysis was performed using SNPtest v2.5.2[39]. Logistic regression was performed assuming an additive model for the allelic effect using the model:

$$\mathrm{male\,pattern\,baldness} \sim \mathrm{age} + \mathrm{pc.1} + \mathrm{pc.2} + \ldots + \mathrm{pc.10} + \mathrm{array\,batch} + \mathrm{genotype}$$

In order to verify if PCs were able to correct for the population structure, we used two different methods applied to the association GWA analysis: genomic control and LD regression. Both methods showed no evidence of test statistic inflation ($\lambda_{GC}$ 1.02, LD regression intercept 1.0021). Finally, in order to verify possible heterogeneity of effect between the two UK Biobank cohorts, we compared effect sizes for the 107 detected distinct SNPs. Figure 1 shows the forest plot for these SNPs for the two UK Biobank populations. No sign of heterogeneity was detected. A total of 3436 cases and 2435 controls were used for this analysis.

For HPFS, men contributing to the current genetic analysis were those previously selected for independent GWAS in nested case–control studies initially designed for a variety of chronic diseases, including type 2 diabetes, coronary heart disease, kidney stone disease, glaucoma, gout, prostate, pancreatic and colon cancer. To allow for maximum efficiency and power, we pooled samples genotyped on the same platforms, which resulted in three data sets herein referred to as HPFS-Affy, HPFS-Illumina and HPFS-Omni (Supplementary Data 1). Detailed methods

and quality assurance pertaining to these genetic data sets have been reported elsewhere (Lindstrom, S. et al. A comprehensive survey of genetic variation in 20,691 subjects from four large cohorts; submitted). Any samples that had substantial (more than one quarter of the distance between the European and non-European centroids on PCA) genetic similarity to non-European reference samples were excluded. Each genetic data set was examined separately and the results were combined in the overall meta-analysis of replication studies. We performed logistic regression assuming an additive genetic model, adjusting for age, initial case–control data set and four principal components of population substructure. A total of 1984 cases and 2857 were used for these analyses. Association analysis was performed only on SNPs required for replication.

ARIC case–control association analysis was performed using logistic regression as implemented by ProbABEL[41]. Additive allelic effects were modelled using the model:

$$\mathrm{male\,pattern\,baldness} \sim \mathrm{age} + \mathrm{pc.1} + \mathrm{pc.2} + \ldots + \mathrm{pc.10} + \mathrm{genotype}$$

A total of 2374 cases and 503 controls were used for the analysis. Association analysis was performed only on SNPs required for replication.

On 23andMe data, logistic regression was performed assuming an additive model for allelic effects, using the model:

$$\mathrm{male\,pattern\,baldness} \sim \mathrm{age} + \mathrm{pc.0} + \mathrm{pc.1} + \mathrm{pc.2} + \mathrm{pc.3} + \mathrm{pc.4} + \mathrm{genotype}$$

This GWA analysis includes data from 9009 cases and 8491 controls of European ancestry, filtered to remove close relatives. Details on sample selection can be found in Pickrell et al.[13]. The results were provided adjusted for a genomic control inflation factor $\lambda = 1.065$. The equivalent inflation factor for 1000 cases and 1000 controls $\lambda_{1000} = 1.007$, and for 10,000, $\lambda_{10,000} = 1.074$.

**Identification of SNPs distinctly associated with MPB and division into loci**. SNPs distinctly associated with the phenotype were estimated using GCTA-COJO[15]. Briefly, this method is comparable to doing a stepwise conditioned regression analysis. The advantage of this approach is that it does not require rerunning of the association analysis, rather it uses the summary statistics from the GWAS and the LD matrix between the SNPs. As a reference panel, we used the genotypes of all the 56,937 genomically British women genotyped in UK Biobank. This choice was due to the attempt to run the analysis also on the X chromosome as if it were an autosome.

Since this last attempt was unsuccessful, direct stepwise conditional analysis was conducted on the X chromosome using PLINK. Briefly, association analysis was rerun limited to the significant SNPs on the whole X chromosome from the discovery step including the top-associated SNP in each of the two loci as covariates. If any genome-wide significant SNP remained in the conditional analysis, the top SNP was included as an additional covariate and association analysis rerun. This procedure was repeated until no other significant SNPs were found. We identified no additional SNPs at locus 70 while 3 distinct SNPs were detected at locus 71.

All replicated SNPs were then divided into loci. To define a locus, we first selected all SNPs with $p$ value $<1 \times 10^{-5}$ and then estimated the distance between each consecutive SNP located on the same chromosome. Two consecutive SNPs were defined to belong to different loci if they were more than 500 kb apart. We considered as independent loci only those which contained at least one distinct SNP. For this reason, two loci on chromosome X were merged with the AR/EDA2R locus.

**Annotation of the distinct SNPs**. Epigenetic signatures were estimated using Haploreg 4.1[20]. As input, we used the 107 distinctly associated SNPs adding to them only those SNPs which were in complete linkage disequilibrium with them, reasoning that they were as good candidates as those selected with the GCTA-COJO analysis. To identify the genes present in each locus, we used DEPICT 1.1 beta[21] which is suitable for use with 1000G phase 3 SNPs. As reference population for the LD pattern, we used the genotypes from all the genomically British women in UK Biobank. Given that, we used DEPICT only to understand which genes were present in each locus and not to run enrichment analysis, as input we used all SNPs which were in the previously defined loci and with a $p$ value $<1 \times 10^{-5}$.

**Heritability estimation**. Heritability was estimated on 20,000 unrelated samples from the discovery cohort using GCTA. Two separate kinship matrices were estimated from SNP array genotype data: one for the autosomes and the other one for chromosome X. We then calculated the heritability explained by each of these matrices using the GCTA-GREML[42] method using the observed prevalence (0.59) to transform the results from the observed to the liability scale. In order to verify how much variance was explained by the identified SNPs, we added the estimated polygenic risk score as a fixed effect in the GREML and measured the decrease in the proportion of variance due to the relationship matrices.

**Gene prioritisation**. Given the particular nature of our phenotype, in order to prioritise the genes present in the 71 identified loci, we created a set of custom criteria. Given the importance of the androgens in MBP, the first criterion was to belong to any of the androgen-related pathways, six genes fulfilled this criterion. The second criterion was if any gene could be linked to hair cycle or growth. We ran the whole

gene list through the ConsensusPathDB-human[22] enrichment software. Nine genes in different loci were annotated as being associated with hair follicle development (GO:0001942) and hair cycle processes (GO:0022405). As a third criterion, we verified if any of the genes in our loci showed significantly different expression patterns between balding and non-balding dermal papilla cell lines at baseline and after stimulation with DHT (the active form of testosterone)[29] (Supplementary Data 21). This analysis revealed 23 differentially expressed genes in 19 loci, 10 at baseline and 13 after stimulation with DHT. As fourth criterion, we used GeneNetwork (http://129.125.135.180:8080/GeneNetwork/) to assess if any of the candidate genes were predicted to be associated with hair cycle or growth. Sixteen additional genes not previously selected were predicted to be related to hair cycle. Fifth, we evaluated the presence of eQTL, which could be explained by the selected SNPs or those in strong LD with them using various approaches: HaploReg v4.1[20], the GENOSCORES platform (see below) and eQTL analysis in peripheral blood. For this last analysis, we used cis-eQTL data from a total of 2360 unrelated individuals obtained from three data sets with gene expression data measured from whole peripheral blood (1240 individuals from Fehrmann-HT12v3, 229 individuals from Fehrmann-H8v2[43] and 891 individuals from the EGCUT study[44] as described. In summary, quality controlled genotype data was imputed using the 1000 Genomes Phase 3 (March 2013 version) cosmopolitan reference panel[45] and imputation dosage values were used for analysis. A more detailed overview of the quality control has been published elsewhere[18]. To detect cis-eQTLs, we assessed only those combinations of SNPs and probes where the distance between SNP and the midpoint of the probe was smaller than 1 megabase (Mb). Individual data sets were meta-analysed using a Z-score method, weighted for the sample size of each data set. The sample labels were permuted (repeated 100 times) in order to obtain the $p$ value distribution used to control the FDR at 5%. Since SNPs can be highly correlated due to LD, cis-eQTL effects are often caused by SNPs in high LD with the disease-associated query SNP. In order to determine whether our disease-associated SNPs have independent cis-eQTL effects with respect to other SNPs in their locus, we performed conditional analysis. Using the procedure described above, we first determined which SNPs show the strongest cis-eQTL (eSNP) effect for each of the probes associated with the 107 disease-associated SNPs (gSNP). Then, we adjusted the gene expression data for these effects using linear regression, and repeated the cis-eQTL analysis on the disease-associated SNPs (and vice versa). This analysis allowed us to identify disease-associated variants that were also the best cis-eQTL SNPs. For the remaining loci, genes were assigned based on the DEPICT gene prioritisation only for those with FDR < 0.05. Supplementary Data 13 summarises for each gene which criteria were met. Druggable genes were annotated using DGI 2.0[23].

**Pleiotropy with previous GWAS**. Pleiotropy with previous GWAS was estimated using the GENOSCORES platform (https://pm2.phs.ed.ac.uk/genoscores/). Briefly, summary results of the GWAS were used to compute genotypic scores for each hit region in the 2432 individuals comprising the UK10K reference panel. These genotypic scores were merged with a table of genotypic scores computed on the same reference panel using a comprehensive database of publicly available GWAS summary statistics for diseases and quantitative traits including gene transcript levels. SNPs were filtered at a $p$ value of $1 \times 10^{-6}$. Each genotypic score was computed as a sum of genotypes (scored as 0, 1, 2 copies of the reference allele) weighted by the effect size estimate (log odds ratio for baldness and other binary traits, regression slope for quantitative traits). For all traits except gene transcript levels, genotypic scores were computed separately for each trait-associated region. For each trait, trait-associated regions were assigned for each genomic region containing at least one SNP with $p$ value $< 1 \times 10^{-7}$. The boundaries of this region were defined by the positions at which there was a gap of at least one megabase from any other SNPs with $p$ value $< 1 \times 10^{-6}$. This procedure yielded 74 regional genotypic scores for baldness, of which 15 were correlated (squared correlation at least 0.5) with at least one other score for another trait. For each block of traits correlated with one of the regional genotypic scores for baldness, the correlations were plotted in a heat map. Results from this analysis were kept only for the 71 significant replicated loci.

**Enrichment analysis of selected genes**. Enrichment analysis was conducted limited to the selected genes using ConsensusPathDB-human[22]. After obtaining the list of enriched pathways and GO terms, we first built an adjacency matrix based on the co-membership of the genes to known pathways. We then used the Luvain[46] method as implemented in the igraph R package to identify subgroups of genes which clustered together (pathway communities). The characteristic that described each community was determined by visually examining how the genes in each community related to the enriched known pathway. We used the same approach using as input the 303-enriched GO terms and then using the Louvain method to identify communities of genes highly interconnected with each other. Given the complexity of the resulting network, it was impossible to assign a significance to each group through visual inspection of the gene-term graph. We thus ran enrichment analysis on each set of genes separately for each identified community. The interaction network resulting from the prioritised genes was created using STRING[28], allowing only for direct interactions.

**Polygenic risk score estimation**. We created two sets of polygenic risk scores plus an overall PGRS comprising all 107 SNPs. Before creating the scores, the betas for

each SNP were re-estimated running logistic regression on all SNPs at the same time using as covariates age and 10 PCs. Genotypes for each SNP were weighted by their beta (dosage × beta). The score at each locus was defined as the sum of all the weighted "distinct" SNPs belonging to the locus: in this way, we obtained 71 locus scores. The overall PGRS was obtained by summing all the locus scores. We then estimated a PGRS for each of the enriched pathways by adding all the locus scores for the loci, which contained the genes in the pathway. "Apoptosis" was merged with "apoptotic signalling pathway", "Wnt signalling pathway" was merged with "Wnt signalling pathway and pluripotency" and "WNT ligand biogenesis and trafficking" with "Wnt signalling in kidney disease" since in all cases the scores were composed of the same loci and were thus identical. We thus obtained 25 further pathway PGRS. Table 2 describes the SNPs used to create each locus score and which loci were attributed to each pathway.

**Association of scores with traits in UK Biobank**. Each polygenic risk score was tested for association with trait using a regression model, with age, sex, genotyping array batch, Townsend deprivation index (a measure of socio-economic status) and the first 15 principal components of the genomic relationship matrix as covariates. Where parental traits were tested, maternal and paternal traits were considered separately. In total, 96 traits were selected for testing for a total of 4320 tests. Storey's q values were used for assessing significance considering multiple testing. Binary traits of participants were tested using logistic regression. The traits tested were smoker, ever smoked and attended college. Disease traits in participants were tested using logistic regression. The traits tested were cancer diagnosis, suffered stroke, suffered depression, diabetic, female reproductive problems, suffered bone fracture, heart disease, multiple sclerosis, peripheral artery disease, psoriasis, renal problems and respiratory problems. Disease traits in parents were tested using logistic regression. The traits tested were Alzheimer's disease, bowel cancer, breast cancer, chronic obstructive pulmonary disease, diabetic, heart disease, hypertension, lung cancer, Parkinson's disease, stroke and prostate cancer. Survival traits of parents were tested using a Cox model with age as the time of survival and alive/dead as status. The trait tested was age.

Quantitative traits in participants were tested using linear regression. The traits tested were educational attainment, height, weight, BMI, body fat %, age at menarche and age at menopause.

**Data availability**. All data that support the findings of this study are available from the corresponding author on reasonable request with the exception of results from 23andMe. To request access to the 23andMe summary statistics, please email apply.research@23andMe.com.

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

## Acknowledgements

This research has been conducted using the UK Biobank Resource under Application Number 8304. We acknowledge funding from the Medical Research Council Human Genetics Unit. We thank the 23andMe research participants and employees for their contributions to this study. We thank the following members of the 23andMe Research Team: Michelle Agee, Babak Alipanahi, Adam Auton, Robert K. Bell, Katarzyna Bryc, Sarah L. Elson, Pierre Fontanillas, Nicholas A. Furlotte, David A. Hinds, Bethann S. Hromatka, Karen E. Huber, Aaron Kleinman, Nadia K. Litterman, Matthew H. McIntyre, Joanna L. Mountain, Elizabeth S. Noblin, Carrie A.M. Northover, Steven J. Pitts, J. Fah Sathirapongsasuti, Olga V. Sazonova, Janie F. Shelton, Suyash Shringarpure, Chao Tian, Joyce Y. Tung, Vladimir Vacic, and Catherine H. Wilson. The ARIC is carried out as a collaborative study supported by National Heart, Lung, and Blood Institute contracts (HHSN268201100005C, HHSN268201100006C, HHSN268201100007C, HHSN268201100008C, HHSN268201100009C, HHSN268201100010C, HHSN268201100011C and HHSN268201100012C), R01HL087641, R01HL59367 and R01HL086694; National Human Genome Research Institute contract U01HG004402; and National Institutes of Health contract HHSN268200625226C. The authors thank the staff and participants of the ARIC study for their important contributions. Infrastructure was partly supported by Grant Number UL1RR025005, a component of the National Institutes of Health and NIH Roadmap for Medical Research. The HPFS is supported by the National Cancer Institute (UM1 CA167552), the National Institute of Diabetes and Digestive and Kidney Diseases (NIDDK, R01DK058845) and the National Heart, Lung, and Blood Institute (NHLBI, R01HL35464), with additional support for the collection and management of genetic data. The HPFS type 2 diabetes (T2D, dbGaP:phs000091.v2.p1) and open-angle glaucoma (GA, dbGaP:phs000308.v1.p1) GWAS were funded as part of the Gene Environment-Association Studies (GENEVA) project under the NIH Genes, Environment, and Health Initiative (T2D: U01HG004399, GA: U01HG004728). Genotyping for the HPFS coronary heart disease GWAS was supported by Merck/Rosetta Research Laboratories, North Wales, PA. The HPFS kidney stone GWAS (dbGaP:phs000460.v1.p1) was supported by NIDDK (5P01DK070756). The HPFS colon cancer GWAS was funded as part of the Colorectal Cancer GWAS Consortium supported by the NCI (U01CA137088 and R01CA059045). T.E. was supported by Estonian Research Council Grant IUT20-60.

## Author contributions

N.P. and J.F.W. wrote the manuscript. N.P., P.K.J., T.E., P.M.M., P.S.d.V. and M.C.C. performed the statistical analyses. A.C.M., N.K., N.F., M.C., E.L.D., A.S., L.F., K.E.N. and P.K. contributed data, tools, resources and funding. All authors critically reviewed the manuscript.

## Additional information

**Competing interests:** The authors declare no competing financial interests.

