## [Peer Review File · Nature Communications]

Reviewers' comments:

Reviewer #1 (Remarks to the Author):

Major claims of the paper

A well-designed and well-executed large-scale GWAS is presented for male pattern-baldness, identifying 71 associated loci, about half of which are new. Identified SNPs are used to estimate heritability and demonstrate high heritability and relatively low genetic complexity, as compared with other common traits investigated with GWAS. The biological relevance of the results is explored by several analyses, including tissue and gene set enrichment. Finally, the data is used to interrogate disease comorbidities previously reported from epidemiological studies, and provides evidence to suggest that comorbidity for different conditions is driven by subsets of MBP risk loci.

Concerns and questions

While the statistical methods and interpretation of the results for the GWAS are appropriate and valid, the approach undertaken for pathway analysis and gene ontology enrichment is not. Genome-wide genetic approaches provide biological insight that is not limited by previous knowledge, and there are many examples in which such studies have revealed disease mechanisms not previously considered, altering our understanding of disease causes and cures. This GWAS identified loci that span more than 200 genes and regulatory elements that could be influencing even more; and then threw away more than half of those genes when attempting to identify and resolve the pathways through which the associations are exerting biological effects.

I have taken some time to repeat pathway analysis and network construction and I feel strongly that they should not prune their list of genes prior to performing pathway analysis, gene ontology enrichment and community identification. The number of genes that are added to these functional groupings is substantial. For example, when I constructed a network with STRING using 200+ genes, the main network consisted of 86 genes and among the most connected genes in the network was one that was omitted from their prioritized list of genes, presumably because it had not been previously linked to regulation of hair growth/cycling. When I repeated GO enrichment and selected significantly enriched pathways that contain at least one WNT gene, the number of WNT-neighbors increased from 21 genes to 46. I think that the functional annotations should be performed on the entire gene set identified by the GWAS, regardless of what has been previously studied. Given the number of genes that are functionally linked in my analyses, I also suspect that the results of the GRS using pathways will also be changed and so should be repeated. All analyses downstream of the association tests should be repeated on the full set of genes and reporting of analysis on their prioritized subset should be removed.

In extended data table 1, all genes that are linked through at least one functional analysis should be listed, in addition to the list of prioritized (i.e. previously studied within the context of hair) gene. This would be instructive of the limitations of relying on previous knowledge.

Minor comments

"Focusing on androgen receptor signalling...five more genes in four loci were identified. " What are these genes and how were they identified? Why wasn't androgen receptor signaling tested in GRS analyses?

"MPB is characterised by a shorter growth (anagen) phase in the hair cycle, which is followed by a catagen phase in which the hair follicle cells go into apoptosis²⁰" This sentence is confusing as written and could be read to suggest that apoptosing cells in catagen is a characteristic of MPB, whereas apoptosis is a characteristic of catagen in healthy HF." MPB is characterized by a shorter growth phase and has been associated with increased apoptosis.

Interpretation of results is problematic for some analyses.

"Although many different pathways besides the androgen receptor pathway have been implicated in the development of MBP²⁰, these results suggest the Wnt and apoptosis pathways play the central roles." Change "these" to "our" to clarify, since the beginning of the sentence is discussing previous knowledge; unless the authors intend considering their findings combined with previous studies. I would remove "the" from central roles. Their analysis did also identify an AR community and AR continues to be the most significant locus and has among the largest effect, so it is not clear that it is less important than apoptosis and WNT signalling.

"35 of the selected genes were connected in a large network with the AR gene at its centre and which also included 8 of the 13 genes differentially expressed in balding dermal papilla cells under DHT stimulation"

When I repeated the analysis, I only found 33 genes connected in a single network. Also, I am not sure how meaningful it is that AR is "at the center", as it only has 3 connections. BCL2 is connected to 10 other genes and 10 other genes are more highly connected to the network than AR. Finally, why do you focus on the DHT stimulated genes, not including genes that are different in the absence of drug?

"the genes in the signalling community (Fig. 2b) are located at the centre of the network and show a high degree of interconnectivity, suggesting considerable interaction amongst the different regulatory systems involved in hair growth. The genes in the nuclear community show less interconnectivity, consistent with their not being in similar pathways. These results suggest that while the genes in the "signalling" community receive the various signals regulating hair growth, they then interact with those in the "nuclear" community to transduce these signals into responses."

An alternative interpretation of the distribution of 'nuclear' genes is that they are less well studied than the 'signaling' genes. This would preclude them both from being assigned to a specific pathway and being well-connected in a STRING graph. TRPS1 is a good example of this, as it has been shown to influence HF biology through WNT signaling, but it is not annotated as such in public databases, perhaps because it has not been as extensively studied. The sentence stating that the results suggest genes in the signaling community

receive signals and interact with the nuclear community to transduce these signals into responses is completely unsubstantiated by the presented data. Furthermore, the validity of their community characterizations as signaling and nuclear is questionable, as there are transcription factors in both communities. The three communities could be labeled as most extensively studied, moderately studied, and least annotated.

Are they novel and will they be of interest to others in the community and the wider field?

Given the tremendous leap forward in the number of loci, this paper will be of interest to investigators interested in hair biology and disorders, as well as those interested in systems to investigate cycles of organ regeneration; i.e. molecular mechanisms engaged to regulate and balance cell growth (wnt) and destruction (apoptosis).

While it has been known for sometime that MPB involves dysregulated wnt signaling and has been previously characterized as having increased markers of apoptosis, this paper is important because of the increased resolution of signaling pathways. For example, WNT signaling influences a wide variety of biological processes in many different contexts, complicating targeting it for any one process. Having increased resolution of mechanism offered here with this GWAS will help to target pathways with precision.

I think that their method for developing pathway specific GRS is innovative and will be of interest and their finding that global assessment misses biological effects is important (Figure 3).

If the conclusions are not original, it would be helpful if you could provide relevant references.

I think that their finding that pathway specific GRS assessment reveals opposing effects is also demonstrated in this paper and should be referenced.
<http://biorxiv.org/content/biorxiv/early/2016/09/23/076794.full.pdf>

Minor edits

In Figure 2 the legend could be clarified by changing the first sentence to indicate that genes in an enriched pathway but not placed in a community are omitted.

I don't understand the difference between supplemental and extended data and the need to be flipping back and forth between two different excel files.

Table S2. Specify the genome build of the reported position coordinates. Indicate sample sizes somewhere, either just in the column header or legend.

Table S3. Histone is misspelled in table description.

Table S5. State the significance threshold in the legend. Also, the genes and loci contributing to the evidence should be listed.

Table S15. It would aid in the interpretation of results if columns were added to indicate the chromosome and bp position for each locus and a second with relevant or an index gene. Also, please explain interpretation of r in the table legend. Does a negative correlation indicate that opposite alleles confer effects?

Extended data T1. For each locus, in addition to my recommendation regarding gene names above, I also recommend adding lowest p and most extreme effect size.

Extended data T5. 'apoptosis' is spelled 'apoptosys'

Reviewer #2 (Remarks to the Author):

In this manuscript, Pirastu and colleagues describe results from a GWAS and replication experiment to identify loci associated with male pattern baldness. This is a highly heritable trait, and already >30 loci have been described, which together explain a quite large fraction of the heritability (for a complex phenotype). The authors found another 35 variants. They then carried out various bioinformatic annotation analyses to prioritize genes, which were then used in enrichment-type analyses. Finally, using the variants within the prioritized genes and pathways, they tested the genetic correlation with other complex human phenotypes. Although doubling the number of MPB loci is important, the secondary in silico analyses remain speculative. I have the following comments:

1. Last sentence of abstract: "Our study not only doubles the number of MPB loci, illuminating the genetic architecture, but also provides a new approach to disentangling the shared biological pathways underpinning complex diseases". Although this sentence is nice, I am wondering if it does not overstate the conclusions of the paper. It was already known that "few" common contribute a lot to MPB risk. The discoveries presented here confirmed this, but did not add anything else (for instance, on the role that rare variants might play in MBP risk). In terms of the underpinning of the MBP biology, this mostly relies on in silico predictions. I would be cautious with this statement as well.

2. The authors should provide more details about the replication cohorts in Table S1. How are non-British ancestry participants from the UKBB sub-divided (African-ancestry, East/South Asian-ancestry, etc.) and what is the sample size of each group? The same question applies to the 23andMe dataset?

3. I am less familiar with HPFS but ARIC includes a large number of African Americans. Why were they excluded from the replication analyses given the use of non-European-ancestry participants from the UKBB and 23andMe?

4. A more important question, related to the 2 previous ones, is how different ethnic groups were analyzed and combined? Among the replicated loci, were there evidence of heterogeneity of effects between ethnicities? And more interestingly, did the authors look

for MPB association signals that were only present in non-Europeans?

5. The following statement in the Methods section is inaccurate: "The two-step approach was used in order to make the analysis time tractable since available software would have taken months to run the analysis even on a large computing cluster." There are several software (we use BOLT-LMM) that can run analyses on the first release of the UKBB within hours (at most a few days). In the two-step approach to analyze the UKBB, is cryptic relatedness accounted for?

6. The first release of the UKBB includes hard genotypes from 2 arrays. Whereas using imputed data is less sensitive to this technical issue, the authors analyzed hard genotypes from the X-chromosome. I could not read how the 2 arrays issue was controlled for in the analysis.

7. Extended Table 2 should include MAF in the discovery cohort. For the replication, although I think MAF should also be reported, it is not so clear to me what is the best way given the strategy to combine individuals from different ancestries. Maybe by ethnicity?

8. I don't understand the genotypic score and pleiotropy analyses. It seems like the authors compute a score for each MBP locus, but it is not clear whether it is only for the sentinel SNPs, the independently associated SNPs, or all the SNPs in a 1Mb region? If it is for all SNPs, how do they handle LD (if at all)? If it is only for a 1 or a few SNPs, how is the test done? Do they look at the correlation one locus at the time, or all loci together (as it is usually done). This section is unclear.

9. In Figure 2, could you use the same color-code for A and B (i.e. red for genes in the AR pathway)? It would be simpler to visualize which genes that belong to the same pathway also physically interact.

10. Analyses of "genetic correlations" between MPB and other complex traits. At the end of this section, I am left wondering how many tests were done and which correlations are actually significant given that number of tests? Are all the results presented in Figure 3 significant after multiple hypotheses correction?

11. In the description of the heritability explained by the autosomes and the X-chromosome, I would like the authors to quantify how much heritability is actually added by the variants identified in this study.

Reviewer #3 (Remarks to the Author):

Pirastu et al. present results from a genome-wide association meta-analysis of male pattern baldness in 25,662 MPB cases and 17,928 controls with replication in an additional 16,824 cases and 14,288 controls. This is the largest study of MPB to date, identifying and replicating 71 loci of which 36 are novel.

General Comments:

The expanded list of replicated loci is impressive. The results are interesting, and I particularly liked the approach looking at pathway-specific genetic correlations.

However, the paper needs polishing. For example, there are two sets of supplementary tables (Supplementary and Extended) which are poorly described (what are the column headers in Table S13?). The text and supplementary tables need checking throughout for typos.

Specific comments:

1. There is very little information on genotyping and quality control of the samples, particularly the discovery analysis in UK Biobank. Genotyping in Biobank was performed using two genotyping arrays (for the UK BiLEVE study and the UK Biobank chip). If samples genotyped using both arrays were included, genotyping array should have been included as a covariate in the discovery analysis.

2. A PCA plot for the non-genomically-British samples should be provided. Do the PCs adequately adjust for the population stratification?

3. What was the genomic control inflation factors for the discovery and replication cohorts? How were these corrected in the analysis?

4. Line 293: Define "substantial".

5. Line 313: "filtered to remove close relatives" is vague.

6. Frequency information should be added for the replicated SNPs in the discovery and replication analysis.

7. While a signal may not be abolished by conditional analysis, it does not mean that it is statistically independent. Signals should be described as distinct rather than independent.

8. The estimate of heritability seems very high. There is no information in the methods to explain how it was calculated or how it compares to previous estimates from the literature.

9. Lines 82-85 (Tissue-specific enhancer 83 enrichment analysis): Placenta is more significant than foreskin keratinocyte?

10. Lines 151-153: GENOSCORE is missing an "S". It would be helpful to have some more detail describing Table S15. Are they all significant?

11. Line 273: "no filter on MAF or imputation quality was applied". In a standard meta-analysis an imputation quality filter would usually be applied. What do the imputation

quality metrics look like for the replication SNPs?

12. Line 322: Why were the genotypes of women, rather than men used?

We would like to thank the reviewers for their comments which we believe have greatly helped in improving the quality of our paper. While the paper was under review a new version of the software used for the enrichment analysis (ConsensusPathDB) was released. Since this is an online only tool it was impossible to access the older version of the software. Furthermore, we wanted to update our results to the latest knowledge, so we have rerun all analyses. Despite this results are very similar to the previous ones. In order to make our responses clearer we have organised the response to the comments of the reviewer in a tabular format which includes the comment of the reviewer, the action we have taken to correct the paper and our response.

Reviewers comment	Actions	Response
Reviewer #1		
I think that the functional annotations should be performed on the entire gene set identified by the GWAS, regardless of what has been previously studied.	We have run the pathways and enrichment analysis on the full list of genes. Added additional table S12 Added relevant reference in the text.	We agree with the reviewer that previous knowledge on genes may limit the ability to detect candidate genes, especially if we consider that we are trying to identify new genes for baldness and thus they would not be described as such. However, it is also true that the increase in biological knowledge allows much better prioritization of the genes in loci and this has become a standard procedure in large GWAS studies. Nevertheless we have run the enrichment analysis on the full set of genes present in our loci. This analysis detected less enriched pathways than with the prioritised genes. In fact, only 9 pathways showed significant enrichment once corrected for multiple testing, while after prioritization the significant pathways increased to 27. Moreover, if we look at the 27 pathways many have been previously functionally linked to hair or baldness before (e.g. TGF-Beta Receptor or Apoptosis). So it seems that by not prioritising we are losing information without any

		apparent gain. Of course previous knowledge may be limiting, however recently there has been a great increase in the information content of resources (i.e. Gtex DB or ENCODE), which diminishes this issue. DEPICT (which prioritises genes based on prior knowledge) is widely used as strong evidence for claiming association. In our case we could not directly use DEPICT given that its annotation files do not include hair tissue, so we had to develop our own criteria, but this does not invalidate the approach as shown by the difference in the enrichment analysis. We have added an additional table with the enrichment analysis results on the whole data set and mentioned the analysis in the text.
"Focusing on androgen receptor signalling...five more genes in four loci were identified. " What are these genes and how were they identified? Why wasn't androgen receptor signaling tested in GRS analyses?	Added additional table to clarify the definition of the AR merged pathway. Revised the legend in table 3 to clarify it.	We are sorry we did not describe this more clearly in the previous version of the manuscript and have added an additional table to clarify this point. The genes were identified by using the list of genes in any of the known Androgen pathways. The genes which fall into this category are those which have "Y" in the "AR merged" column in table 3. None of the AR pathway was tested with the PGRS analysis because this was limited to the significantly enriched pathways and none of the androgen-related pathways was amongst them.
"MPB is characterised by a shorter growth (anagen) phase in the hair cycle, which is followed	Corrected the text according to the reviewer's suggestion.	We agree with the reviewer that the previous formulation was imprecise and have corrected

by a catagen phase in which the hair follicle cells go into apoptosis²⁰ This sentence is confusing as written and could be read to suggest that apoptosing cells in catagen is a characteristic of MPB, whereas apoptosis is a characteristic of catagen in healthy HF." MPB is characterized by a shorter growth phase and has been associated with increased apoptosis.		the text accordingly.
"Although many different pathways besides the androgen receptor pathway have been implicated in the development of MBP20, these results suggest the Wnt and apoptosis pathways play the central roles." Change "these" to "our" to clarify, since the beginning of the sentence is discussing previous knowledge; unless the authors intend considering their findings combined with previous studies. I would remove "the" from central roles. Their analysis did also identify an AR community and AR continues to be the most significant locus and has among the largest effect, so it is not clear that it is less important than apoptosis and WNT signalling.	We have rephrased the sentence to clarify this issue.	Again we agree with the reviewer that the previous version of the paper was not clear on this point. We have corrected the text to clarify that AR is still the most important gene and that Wnt signalling and apoptosis are important beside the AR pathway.
"35 of the selected genes were connected in a large network with the AR gene at its centre and which also included 8 of the 13 genes differentially expressed in balding dermal papilla cells under DHT stimulation" When I repeated the analysis, I only found 33 genes connected in a single network. Also, I am not sure how meaningful it is that AR is "at the center", as it only has 3 connections. BCL2 is connected to 10 other genes and 10 other genes are more highly connected to the network than AR. Finally, why do you focus on the DHT stimulated genes, not including genes that are different in the absence of drug?	Corrected the text.	We agree with the reviewer, the number was wrong and referred to an intermediate version of the network, it is now correct. We did not focus on DHT-stimulated genes, the initial list included all selected genes (i.e. differentially expressed with or without DHT), however those differentially expressed under DHT stimulation were mostly retained in the network. We have removed the sentence referring to AR as the center of the network.
"the genes in the signalling community (Fig. 2b) are located at the centre of the network and show	Changed text and figures to reflect new	We agree with the reviewer that the difference

a high degree of interconnectivity, suggesting considerable interaction amongst the different regulatory systems involved in hair growth. The genes in the nuclear community show less interconnectivity, consistent with their not being in similar pathways. These results suggest that while the genes in the “signalling” community receive the various signals regulating hair growth, they then interact with those in the “nuclear” community to transduce these signals into responses.” An alternative interpretation of the distribution of ‘nuclear’ genes is that they are less well studied than the ‘signaling’ genes. This would preclude them both from being assigned to a specific pathway and being well-connected in a STRING graph. TRPS1 is a good example of this, as it has been shown to influence HF biology through WNT signaling, but it is not annotated as such in public databases, perhaps because it has not been as extensively studied. The sentence stating that the results suggest genes in the signaling community receive signals and interact with the nuclear community to transduce these signals into responses is completely unsubstantiated by the presented data. Furthermore, the validity of their community characterizations as signaling and nuclear is questionable, as there are transcription factors in both communities. The three communities could be labeled as most extensively studied, moderately studied, and least annotated.	results.	in connectivity could be due to a difference in our knowledge of the involved genes. To verify this we compared the number of GO terms associated to each gene in each community which should reflect our knowledge of each gene. The median number of GO terms was much higher (65.5 vs 40) in the “other” community which would suggest that the clustering is not due to a difference in knowledge but by actual shared functions. Please note that given the change in the enrichment software version we have rerun the whole analysis. The communities detected are now 2, 1 is still characterised by signalling pathways and the other by pathways related to development. We have changed the figure and text accordingly.
“Focusing on androgen receptor signalling...five more genes in four loci were identified. “ What are these genes and how were they identified? Why wasn’t androgen receptor signaling tested in GRS analyses?	Corrected the text	We merged of all the pathways in which the androgen receptor is involved to generate the androgen receptor broad pathway. To clarify this we have added table S17 which describes the pathways used and the genes involved. This “merged” pathway was not included in the PGRS analysis because none of its component pathways were significant in the

		enrichment analysis.
In Figure 2 the legend could be clarified by changing the first sentence to indicate that genes in an enriched pathway but not placed in a community are omitted.	Clarified here	All genes which were in enriched pathways were included in figure 2. Some of the genes are missing because they are not in the enrichment analysis. Community detection assigns all genes in a network to a community so all of them are included in one group or another.
I don't understand the difference between supplemental and extended data and the need to be flipping back and forth between two different excel files.	We have included the extended data tables as main display items.	We are sorry for the confusion, this is due to the transfer from a different journal of the Nature group which allowed only 3 display items and thus all tables were included as "extended data". The tables have now been included in the main file at the end of the paper.
Table S15. It would aid in the interpretation of results if columns were added to indicate the chromosome and bp position for each locus and a second with relevant or an index gene. Also, please explain interpretation of r in the table legend. Does a negative correlation indicate that opposite alleles confer effects?	Added columns with positions	We agree with the reviewer and have added the columns with the position and boundaries of the loci. The negative correlation indicates that the effects for each allele are opposite as compared to baldness, for example, at locus 13 a higher predisposition to baldness correlates with a lower red blood cell count.
Extended data T1. For each locus, in addition to my recommendation regarding gene names above, I also recommend adding lowest p and most extreme effect size.	Added smallest p-value column to the table.	We agree with the reviewer and have added the smallest p-value for each locus. Regarding the effect size we did not add the column because the biggest effect (beta) does not always coincide with the smallest p-value because of the MAF and thus would not refer to the same SNP, possibly creating confusion.
Reviewer #2		
1. Last sentence of abstract: "Our study not only	Clarified here	We do not feel that the central claims of this

doubles the number of MPB loci, illuminating the genetic architecture, but also provides a new approach to disentangling the shared biological pathways underpinning complex diseases". Although this sentence is nice, I am wondering if it does not overstate the conclusions of the paper. It was already known that "few" common contribute a lot to MPB risk. The discoveries presented here confirmed this, but did not add anything else (for instance, on the role that rare variants might play in MBP risk). In terms of the underpinning of the MBP biology, this mostly relies on in silico predictions. I would be cautious with this statement as well.		sentence are overstated. Firstly we did double the number of MPB-associated loci. Secondly, the various analyses do shine more light on the genetic architecture – we didn't claim to explain or understand it all, just that the new genes illuminated it. Finally we do present a new approach – pathway-specific genetic correlations – to disentangle shared effects. We agree that we did not explore rare variants but we did not claim that.
2. The authors should provide more details about the replication cohorts in Table S1. How are non-British ancestry participants from the UKBB subdivided (African-ancestry, East/South Asian-ancestry, etc.) and what is the sample size of each group? The same question applies to the 23andMe dataset? 3. I am less familiar with HPFS but ARIC includes a large number of African Americans. Why were they excluded from the replication analyses given the use of non-European-ancestry participants from the UKBB and 23andMe?	We have added lambda and LD regression intercept as means of checking for pvalue inflation in the UK Biobank replication cohort.	We agree with the reviewer that the details about the cohorts were not completely clear and have added information about ethnicity in the paragraphs which describe each cohort. All samples used in the replication cohorts are of European descent. In the case of ARIC African Americans were excluded because the number of cases and controls were too few to create a group on its own. In UK Biobank analyses, only subjects who self identified as "British" we used. For discovery, the subset of those that UKBiobank themselves had determined as also genomically british (using stringent PC clustering) was used. For replication the remaining self-identified British samples were used. In the analysis of the self reported British "non genetically British" group, PCs are able to properly correct for potential stratification as shown both by genomic control and LD regression intercept which show no evidence

		of p-value inflation in the analyses.
4. A more important question, related to the 2 previous ones, is how different ethnic groups were analyzed and combined? Among the replicated loci, were there evidence of heterogeneity of effects between ethnicities? And more interestingly, did the authors look for MPB association signals that were only present in non-Europeans?	Clarified here	Almost all samples used were of European descent. In the case of the UK biobank “other” cohort used there is some underlying structure to the whole population which was adjusted for by using the PC as covariates. Given that the vast majority of samples were of European descent, we were unable to perform ethnicity-specific analyses.
5. The following statement in the Methods section is inaccurate: “The two-step approach was used in order to make the analysis time tractable since available software would have taken months to run the analysis even on a large computing cluster.” There are several software (we use BOLT-LMM) that can run analyses on the first release of the UKBB within hours (at most a few days). In the two-step approach to analyze the UKBB, is cryptic relatedness accounted for?	Clarified the text	We are sorry we were not clear before and changed the text to clarify this issue. The two-step approach was necessary to get the proper logistic model association analysis. In fact, although there are many softwares which perform linear regression analysis in a time efficient way, none of them will perform logistic regression analysis which is needed to get betas on the correct scale. For this reason we used the linear regression as a first scan and repeated the analysis with the correct model only on the SNPs which were significant.
6. The first release of the UKBB includes hard genotypes from 2 arrays. Whereas using imputed data is less sensitive to this technical issue, the authors analyzed hard genotypes from the X-chromosome. I could not read how the 2 arrays issue was controlled for in the analysis.	Clarified the text	We agree with the reviewer that this is an important point for the analysis plan and are sorry we did omit it from the previous version of the paper. The array difference was checked by using array batch as a covariate in the analysis which implicitly corrects also for differences in arrays. We have now specified this in the text.
7. Extended Table 2 should include MAF in the discovery cohort. For the replication, although I think MAF should also be reported, it is not so clear to me what is the best way given the	Added to the table	We agree with the reviewer and have added the Discovery cohort MAF to the table. Given that the table will be in the main text it

strategy to combine individuals from different ancestries. Maybe by ethnicity?		is not possible to add MAF also for the replication cohorts. We have added this information in the supplementary table S2.
8. I don't understand the genotypic score and pleiotropy analyses. It seems like the authors compute a score for each MBP locus, but it is not clear whether it is only for the sentinel SNPs, the independently associated SNPs, or all the SNPs in a 1Mb region? If it is for all SNPs, how do they handle LD (if at all)? If it is only for a 1 or a few SNPs, how is the test done? Do they look at the correlation one locus at the time, or all loci together (as it is usually done). This section is unclear.	Clarified the text.	We are sorry we did not describe this properly in the previous version of the paper we have thus reviewed the text to reflect better what was done. For each locus we used only the "independent" SNPs present in the specific locus. The Genoscore the method is different: it is based on calculating the scores in the region and then using the resulting PGRS to correlate with the other scores on a reference population. In this way it is possible to see correlations, even if the SNPs used are not exactly the same.
9. In Figure 2, could you use the same color-code for A and B (i.e. red for genes in the AR pathway)? It would be simpler to visualize which genes that belong to the same pathway also physically interact.	Clarified here	We are sorry this is unclear from the legend and have reviewed it to clarify it. We used 2 different colour codes because the colours indicate different community detections. In the first the "clusters" were created based on the co-membership of common known pathways while the second is based on shared GO terms. For this reason using the same colour coding would risk confusing the reader and thus this choice.
10. Analyses of "genetic correlations" between MPB and other complex traits. At the end of this section, I am left wondering how many tests were done and which correlations are actually significant given that number of tests? Are all the results presented in Figure 3 significant after	Clarified the text	We have added the number of tests and multiple test correction to the text. The results in figure 3 are all significant at $FDR < 0.05$ using Storey's q-values.

multiple hypotheses correction?		
11. In the description of the heritability explained by the autosomes and the X-chromosome, I would like the authors to quantify how much heritability is actually added by the variants identified in this study.	Added information to the text.	We have estimated the proportion of heritability explained by the previous 12 known and replicated loci and added the information to the text. In particular, the previously described loci explained only about 9% of the autosomal heritability and 56% of the X chromosome's. We have increased this proportion greatly, especially that due to the autosomal variants.
Reviewer #3		
1. There is very little information on genotyping and quality control of the samples, particularly the discovery analysis in UK Biobank. Genotyping in Biobank was performed using two genotyping arrays (for the UK BiLEVE study and the UK Biobank chip). If samples genotyped using both arrays were included, genotyping array should have been included as a covariate in the discovery analysis.	Clarified the text	We agree that we have not been clear on this point before and have corrected the text accordingly. 96% of the SNPs overlap between the two arrays, but we used array batch as a covariate in all UK Biobank analyses which implicitly also corrects the analysis for differences in the genotyping array.
2. A PCA plot for the non-genomically-British samples should be provided. Do the PCs adequately adjust for the population stratification?	Added information to the text	We agree with the reviewer that this is an important concern. We have thus extended the association analysis for this and estimated both lambda GC and the intercept from the LD regression. Both measures agree that PCs are properly correcting possible biases due to the population structure.
3. What was the genomic control inflation factors for the discovery and replication cohorts? How were these corrected in the analysis?	Added the analysis details of UK biobank in the Methods section	Initially inflation was controlled only in the discovery cohort, because we were otherwise seeking only replication. We have however extended the analysis for the UK Biobank

		replication cohort to demonstrate that the population genetic structure is well controlled.. For the other cohorts only the replication SNPs were required and thus no control was applied.
4. Line 293: Define "substantial".	Clarified here	More than one quarter of the distance between the European and non-European centroids on PCA.
5. Line 313: "filtered to remove close relatives" is vague.	Clarified here	
7. While a signal may not be abolished by conditional analysis, it does not mean that it is statistically independent. Signals should be described as distinct rather than independent.	Corrected the text	
8. The estimate of heritability seems very high. There is no information in the methods to explain how it was calculated or how it compares to previous estimates from the literature.	Added a paragraph in the text to explain how it was obtained	We agree with the reviewer that we did not describe the method and have thus added a specific paragraph. The heritability estimates were calculated on the UK Biobank genomically British cohort taking 20 thousand random individuals and using the GCTA GRM (genetic relationship matrix) method. All people who had at least one relative in the UK Biobank cohort were excluded prior to the selection of the sample. We used 2 separate kinship matrices (1 for the autosomes and the other for the X chromosome) to partition the heritability. We also agree with the reviewer that the heritability is extremely high. However, this is not surprising given previous estimates from twins. Our estimate of heritability is slightly

		higher than that from twin studies. This is in part because we use a case-control approach while previous twin heritability studies have actually used the grade of baldness as a quantitative trait, which introduces noise from the determination of the correct grade.
9. Lines 82-85 (Tissue-specific enhancer 83 enrichment analysis): Placenta is more significant than foreskin keratinocyte?	Clarified here	Unfortunately the correct tissue (hair bulge follicle cells) is not present in ENCODE. This is a great limitation to any tissue-specific analysis, given that baldness is not only specific to a tissue but also it specifically shows a different pattern across the scalp. The effects of androgens are specific to different areas of skin since the same hormone in a different body part makes hair grow (e.g. beard hair). This is possibly the reason for these unusual results.
10. Lines 151-153: GENOSCORE is missing an "S". It would be helpful to have some more detail describing Table S15. Are they all significant?	Clarified here	We have fixed the typo. P-values have not been estimated because they would all be significant given that the estimates used over 10 thousand people. With such numbers even correlations of 1% would be highly significant, for this reason we reported only extremely strong correlations which seem more meaningful than small p-values.
11. Line 273: "no filter on MAF or imputation quality was applied". In a standard meta-analysis an imputation quality filter would usually be applied. What do the imputation quality metrics look like for the replication SNPs?		We did not apply any filter to the replication analysis because we had a very large sample size and were looking at SNPs which had already given a significant positive signal in the discovery cohort. This approach is conservative considering that low info scores will generate false negatives and not false positives;

		however we retain the possibility to replicate findings.
12. Line 322: Why were the genotypes of women, rather than men used?	Clarified here	This choice was due to the fact that GCTA-COJO works best with an as large as possible a sample size. We wanted to avoid using the same samples from the discovery or replication cohorts and since most UK Biobank men were in one or other, we opted to use all women. The genotypes are used only to derive the LD pattern, so sex does not matter for the analysis on the autosomes. For the X chromosome a different method was used (as specified in the materials and methods section).

Reviewers' comments:

Reviewer #1 (Remarks to the Author):

The authors have done an excellent job of responding to the original review and I think that the revised manuscript is much improved. I have editorial suggestions in the manuscript pdf, which I will upload. In addition, the text in Figure 2 displays will need to be enlarged, it is not legible as it currently is.

The only additional concern that I have is reconciling this paper with the two MPB papers that have been published in 2017: (1) Hagenaars et al uses the same cohort and while the goal of that project is to identify a classifier, a GWAS is first conducted and associations are reported and largely overlap with those reported here; (2) Heilmann-Heimbach et al uses a discovery and replication cohort that I think are largely distinct from those used here, with the exception of 23&me, but that should be clarified. There are some differences in identified loci among these three publications. I think that this current paper still presents novel findings, in particular the extensive work in identifying distinct processes and differentially linking them to subsets of comorbidities/co-conditions.

Reviewer #2 (Remarks to the Author):

The authors have addressed appropriately my concerns. I don't have additional comments.

Reviewer #3 (Remarks to the Author):

Pirastu et al. have not addressed many of my comments and many details are lacking in the paper. It would be helpful to have changes in the text highlighted, and clearly described in the response to reviewers.

1. Information on genotyping and quality control. Although the authors have clarified that they adjusted for array batch in the analysis, no additional details on the genotyping and quality control of the data have been added. Did they use the QC filters provided by Biobank or did they apply additional filters? Given this is the discovery analysis it should be clear how they treated the data. Why have the equations for the regression been presented for the replication cohorts, but not the discovery sample?

2. Lack of inflation in the non-genomically-British samples could also be lack of power. Do the authors see the same results when running a linear mixed model? Following on from reviewer 2's comment, was there any evidence of heterogeneity for the variants in the non-genomically-British samples? It would be useful to provide forest plots for the novel loci.

3. Genomic control. I can't see the details of the genomic-control inflation factors, and how these were corrected for in the analysis apart from in the non-genomically-British UK Biobank samples. Please provide for the discovery and for the other replication cohorts

where genome-wide data are available. If only the replication SNPs were requested/provided that should be made clearer in the text.

4. Line 305. "substantial" This needs to be clarified in the text as well!

5. Line 325 (previously 313). This comment has not been addressed in the text or in the response to reviewers. You should clarify exactly what this means in the text.

6. Distinct vs. independent. While some instances have been corrected, the authors still describe the "Identification of SNPs independently association with MPB". Similarly Table 2 has distinct in the title, but independent in the legend. Distinct does not need to be in quotes.

7. Heritability. Again, the details are lacking in the methods. Were the kinship matrices estimated from the directly genotyped or imputed data? What estimate of prevalence was used? (as this can have a substantial impact on the estimate of heritability). There is still no mention of how this compares with previous estimates in the text.

8. Line 158. GENSCORE is still missing an S.

9. Genotypes of women. Surely you would want to use an LD reference panel that most closely resembles the discovery sample? In fact, if the genotypes are available (which they are), you should use the discovery sample set
<http://gcta.freeforums.net/thread/178/conditional-joint-analysis-using-summary>

We would like to thank the reviewers for their comments which we believe have greatly helped in improving the quality of our paper. In order to make our responses clearer we have organised the response to the comments of the reviewer in a tabular format which includes the comment of the reviewer, the action we have taken to correct the paper and our response.

Reviewers comment	Actions	Response
Reviewer #1		
The authors have done an excellent job of responding to the original review and I think that the revised manuscript is much improved. I have editorial suggestions in the manuscript pdf, which I will upload. In addition, the text in Figure 2 displays will need to be enlarged, it is not legible as it currently is.	Provided separate figure files	We have incorporated all editorial suggestions in the text. The only exception is in changing the “distinct signals” into independent as it clashes with reviewer #3’s requirements. We agree with the reviewer about the figure; the illegibility is due to including the figure in the word document. We have provided separate high definition figure files.
The only additional concern that I have is reconciling this paper with the two MPB papers that have been published in 2017: (1) Hagenaars et al uses the same cohort and while the goal of that project is to identify a classifier, a GWAS is first conducted and associations are reported and largely overlap with those reported here; (2) Heilmann-Heimbach et al uses a discovery and replication cohort that I think are largely distinct from those used here, with the exception of 23&me, but that should be clarified. There are some differences in identified loci among these three publications. I think that this current paper still presents novel findings, in particular the extensive work in identifying distinct processes and differentially linking them to subsets of comorbidities/co-conditions.	Corrected the text throughout to reflect the new number of loci. Eliminated the “known” column from table 1. Added comment on the Haagenars paper (lines 162-169). Added references to the Heilmann-Heimbach et al paper.	We have corrected table 1 and the Manhattan plot to reflect the new findings reported while this work was in review. We have thus corrected the number of novel loci to reflect the overlap with Heilmann-Heimbach et al. (from 36 to 30). Looking at the Hagenaars paper we cannot consider their findings as such a properly replicated genetic association study. In fact, given that their goal was to create a predictor for MBP, they did not use any replication cohort, thus they also report 3 loci which are shown to be false positives when we sought replication. The remaining 18 loci are below significance threshold in our

		analysis. This could be also due to a different definition of the phenotype as they have included as cases people in “category II” in UK biobank. This in our opinion is not correct, given that it includes also people with a II grade MBP on the Norwood–Hamilton scale, which is highly confounded. The fact that this phenotype is “poorer” is shown by the much weaker association they recovered for the X-Chromosome locus compared to our analysis. We have thus reported their study in the text but did not consider their findings as replicated discoveries.
Reviewer #3		
Pirastu et al. have not addressed many of my comments and many details are lacking in the paper. It would be helpful to have changes in the text highlighted, and clearly described in the response to reviewers.	We have clarified all the points made.	
Information on genotyping and quality control. Although the authors have clarified that they adjusted for array batch in the analysis, no additional details on the genotyping and quality control of the data have been added. Did they use the QC filters provided by Biobank or did they apply additional filters? Given this is the discovery analysis it should be clear how they treated the data. Why have the equations for the regression been presented for the replication	Added additional information to the text to clarify these points. (see lines 190 to 195)	We did not run any additional QC to the one provided by UK biobank.

cohorts, but not the discovery sample?		
Lack of inflation in the non-genomically-British samples could also be lack of power. Do the authors see the same results when running a linear mixed model? Following on from reviewer 2's comment, was there any evidence of heterogeneity for the variants in the non-genomically-British samples? It would be useful to provide forest plots for the novel loci.	Provided forest plot as additional figure and specified the text. (see additional figure 1 and text lines 276 to 279	We understand the proper concern on population stratification and we have excluded its possible presence with genomic control and LD regression. In particular this last approach is able to detect modest stratification even in sample sizes much smaller than the one used in our study (Bulik-Sullivan 2015) so it seems highly unlikely that our failure to detect p-value inflation is due to a lack of power. Given the lack of any sign of evident stratification it seems unreasonable to run a linear mixed model. We did not detect any sign of heterogeneity of effects between the genomically and non-genomically british samples, we have included an additional figure comparing the effects and standard errors in the two populations (Additional Figure 1).
Genomic control. I can't see the details of the genomic-control inflation factors, and how these were corrected for in the analysis apart from in the non-genomically-British UK Biobank samples. Please provide for the discovery and for the other replication cohorts where genome-wide data are available. If only the replication SNPs were requested/provided that should be made clearer in the text.	Added information to the text for lambda estimation (lines 252 to 254 for UK biobank discovery cohort and 315 to 318 for 23andMe). For the other cohorts: HPFS lines 297-298 and ARIC lines 306-307.	We have added the lambda estimated with GC and LD regression also for the discovery step. We have specified that only replication SNPs were required from the collaborating cohorts.

Line 305. "substantial" This needs to be clarified in the text as well!	We have corrected the text to reflect the reviewer's concerns (lines 292-293)	
Line 325 (previously 313). This comment has not been addressed in the text or in the response to reviewers. You should clarify exactly what this means in the text.	We have corrected the text to reflect the reviewer's concerns (lines 315-316)	
Distinct vs. independent. While some instances have been corrected, the authors still describe the "Identification of SNPs independently association with MPB". Similarly Table 2 has distinct in the title, but independent in the legend. Distinct does not need to be in quotes.	We have changed all references to independently to distinct removing the quotes.	
Heritability. Again, the details are lacking in the methods. Were the kinship matrices estimated from the directly genotyped or imputed data? What estimate of prevalence was used? (as this can have a substantial impact on the estimate of heritability). There is still no mention of how this compares with previous estimates in the text.	We have added information to the text to reflect these points. (lines 359, 361-363, 364-365)	We specified that the kinship matrix had been estimated from the genotyped data. As prevalence we used the observed one given that UK biobank is probably the largest population cohort with MBP phenotype. In the literature, prevalence ranges between 0.5 to 1. Given that our estimate (0.589571) is close to the lower bound, heritability is unlikely to be overestimated, since higher prevalence values would have given even higher estimates. Our estimates are in line with those previously observed when MBP was analysed as a binary trait. We have added comparisons to the previous estimates for MBP in the text.
Line 158. GENOSCORE is still missing an S.	Addes the S to GENOSCORES	

Genotypes of women. Surely you would want to use an LD reference panel that most closely resembles the discovery sample? In fact, if the genotypes are available (which they are), you should use the discovery sample set http://gcta.freeforums.net/thread/178/conditional-joint-analysis-using-summary	Added further information in Materials and Methods section. (lines 328-332)	Unfortunately GCTA-COJO does not work on the X chromosome. We used the women in order to perform GCTA analysis of the X chromosome as though it were an autosome. This did not work and thus we used actual conditional analysis on the X chromosome. Using the women from the same population does not affect the GCTA analysis as it uses the LD patterns amongst SNPs which we are sure are the same in men and women on the autosomes. Thus rerunning the analyses seems unjustified as it would give an almost identical result.
---	--	--

REVIEWERS' COMMENTS:

Reviewer #3 (Remarks to the Author):

The revised version is clearer and the authors have addressed my comments.

Minor comments:

Legend for Additional Figure 1 should clarify it is only for the genomically/non-genomically British samples from UK Biobank. Suggest adding "(in green)" after "Non-Genomically British". Genomically doesn't need a capital letter.

Numbers in Table 2 should be rounded appropriately (suggest 3 decimal places for Beta, SE and MAF).

Similarly suggest rounding numbers in Tables 3 and 7.

Line 292: "More" should have a lowercase "m"

Line 329 "as if it were an" (suggest adding an "if")